# Enhanced electrophysiological recordings in acute brain slices, spheroids, and organoids using 3D high-density multielectrode arrays

Lisa Mapelli[1]*, Danila Di Domenico[1], Giacomo Sciacca[2], Francesco Mainardi[1], Alessandra Ottaviani[1], Anita Monteverdi[3], Mariateresa Tedesco[2], Chiara Rosa Battaglia[2], Simona Tritto[1], Mauro Gandolfo[2], Kilian Imfeld[2], Stefanie Kiderlen[4], Lukas Krainer[4], Chiara Cervetto[5,6], Manuela Marcoli[5,6,7], Anson Sing[8], Jimena Andersen[8,9], Fikri Birey[8,9], Steven A. Sloan[8,9], Alessandro Maccione[2☯], Egidio D'Angelo[1,3☯]

1 Department of Brain and Behavioral Sciences, University of Pavia, Pavia, Italy, 2 Discovery Lab, 3Brain AG, Pfäffikon, Switzerland, 3 Digital Neuroscience Center, IRCCS Mondino Foundation, Pavia, Italy, 4 Contract Imaging Service, Prospective Instruments LK GmbH Co KG, Dornbirn, Austria, 5 Department of Pharmacology (DIFAR), University of Genoa, Genoa, Italy, 6 Interuniversity Center for the Promotion of the 3Rs Principles in Teaching and Research (Centro 3R), Genoa, Italy, 7 Center of Excellence for Biomedical Research, Genoa, Italy, 8 Department of Human Genetics, Emory Brain Organoid Hub, Atlanta, Georgia, United States of America, 9 Department of Human Genetics, Emory University School of Medicine, Atlanta, Georgia, United States of America,

☯ These authors contributed equally to this work.
* lisa.mapelli@unipv.it

## Abstract

Recent advances in three-dimensional (3D) biological brain models *in vitro* and *ex vivo* are creating new opportunities to understand the complexity of neural networks but pose the technological challenge of obtaining high-throughput recordings of electrical activity from multiple sites in 3D at high spatiotemporal resolution. This cannot be achieved using planar multi-electrode arrays (MEAs), which contact just one side of the neural structure. Moreover, the specimen adhesion to planar MEAs limits fluid perfusion along with tissue viability and drug application. Here, the efficiency of the tissue-sensor interface provided by advanced 3D high-density (HD)-MEA technology was evaluated in acute brain slices, spheroids, and organoids obtained from different brain regions. The 3D HD-MEA microneedles reached the inner layers of samples without damaging network integrity and the microchannel network between microneedles improved tissue vitality and chemical compound diffusion. In acute cortico-hippocampal and cerebellar slices, signal recording and stimulation efficiency proved higher with the 3D HD-MEA than with a planar MEA improving the characterization of network activity and functional connectivity. The 3D HD-MEA also resolved the challenge of recording from brain spheroids as well as cortical and spinal organoids. Our results show that 3D HD-MEA technology represents a valuable tool to address the complex spatiotemporal organization of activity in brain microcircuits, making it possible to investigate 3D biological models.

**Data availability statement:** All data files are available at the following link: https://doi.org/10.5281/zenodo.16028942.

**Funding:** The authors acknowledge the #NEXTGENERATIONEU (NGEU) and Ministry of University and Research (MUR), National Recovery and Resilience Plan (NRRP), project MNESYS (PE0000006) – A Multiscale integrated approach to the study of the nervous system in health and disease (DN. 1553 11.10.2022) to ED. This project has received funding from the European Union's Horizon 2020 research and innovation programme under the grant agreement 964877 – NEUCHIP to AMa. The authors acknowledge the European Union – Next GenerationEU – National Recovery and Resilience Plan (NRRP) – Mission 4 Component 2 Investiment 1.1 Call PRIN 2022 PNRR CUP F53D23010320001 (MUR code P2022YMM29 - Functional and computational investigation of brain networks in PCDH19-related developmental and epileptic encephalopathy-9. A close-up on Parvalbumin interneurons) to LM. 3Brain AG provided support for this study in the form of salaries to MT, GS, CRB, MG, KI, and AM. The specific roles of these authors are articulated in the 'author contributions' section. The sponsors did not play any role in the design, data collection, analysis, decision to publish, or preparation of the manuscript.

## Introduction

A variety of human and animal-derived 3D models such as tissue slices, spheroids, organoids, and multicellular 3D cultures are available today to recapitulate critical features of brain structure and function [1] with higher accuracy and relevance compared to classical 2D cultures [2]. However, the increasing complexity of these 3D systems raises new methodological challenges. From a physiological point of view, a critical issue is probing the interconnected inner layers of these structures with enough spatiotemporal resolution to accurately describe their complex network activity with minimal perturbation of the biological system [3].

One of the most promising approaches is represented by multi-electrode arrays (MEAs), cell-electronic biointerfaces widely used to perform non-invasive, label-free, and multisite extracellular recordings of brain activity *in vitro* and *ex vivo*. Other than on primary or induced Pluripotent Stem Cell (iPSC) derived 2D cultures [4–7], these devices are also employed on structured 3D models as acute brain slices and explanted tissues [8–14] and in 3D culturing models like spheroids and organoids [15–17]. However, several critical limitations arise when using this technology for 3D models.

A major issue is that most commercial and custom-made MEAs are planar, resulting in recorded signals that primarily originate from the outermost layers, with minimal or no access to the activity of the cells inside the 3D structure. This is even more relevant in acute slices [18] for the presence of a layer of dead cells caused by the cutting procedure, and in organoids [19] where a scaffold matrix (e.g., Matrigel) often encapsulates the tissue (which might also provide insulation). Several approaches have been developed in the MEA field to produce penetrating electrodes capable of retrieving signals from the bulk of the 3D model, thus overcoming this main issue. The first attempts to manufacture 3D electrodes date back to the early '90s by Hoogerwerf et al. [20] and similar approaches were proposed in the following decades [21–30]. Alternative approaches are represented by MEAs composed of flexible electrodes surrounding the surface of the 3D sample [19] or made of a stretchable mesh that can integrate into the 3D sample during its growth [31].

A second issue in measuring 3D models activity is the need to record from many sites over large areas to precisely capture the complex and intricate activity of 3D tissues [32,33]. A partial solution was provided by the adoption of microchip-based High-Density MEAs (HD-MEAs), which integrate thousands of electrodes with cellular or subcellular spatial resolution to provide a detailed spatiotemporal map of functional activity [2,34–36]. A recent approach to combine HD-MEA with 3D electrode technologies was proposed by Wang et al. [37]. This method is currently optimized and tested for retinal applications [38].

A third issue is to cope with the high cellular density and spatial arrangement of 3D models, which require efficient nutrient diffusion, proper oxygenation, and fast metabolic waste removal to avoid rapid necrosis of the core of the tissue [39]. Conventional or HD-MEA devices require cell-electrode intimate contact to acquire signals. This implies that the tissue must adhere well to the surface containing the electrodes,

thus strongly limiting the solution exchange at the recorded layers. This often leads to a rapid decrease in activity and poor physiological conditions. To address this specific issue, ad hoc solutions such as those proposed by Killian et al. [40] and perforated MEAs [41] have been developed.

In 2023, a novel and advanced generation of 3D HD-MEAs was introduced to the scientific community to address the complex challenges of interfacing with 3D tissue. Building on microchip-based MEA technology already validated on several biological models [42–46], such 3D HD-MEAs provide thousands of bidirectional (for recording and stimulation) microneedles (µneedles) with pedestals at their base, forming microchannels when interfaced with the tissue, thus avoiding complete contact with the chip base. This results in a 3D HD-MEA with a 64 x 64 µneedle electrodes grid with an integrated microfluidic system for oxygen, nutrient, and chemical diffusion at the bottom layers of the tissue. This technology represents the first attempt to integrate penetrating capability, high spatiotemporal resolution recording, targeted cell stimulation, and improved tissue viability.

In this paper, we exploited this novel 3D HD-MEA technology to perform electrophysiological recordings from 3D biological preparations including acute brain slices, spheroids, and organoids, which are typically hard to deal with on planar chips [47,48]. We first demonstrated microneedles' penetration into the tissue and microchannels' effectiveness in improving tissue viability. Then, we evaluated 3D HD-MEA effectiveness in detecting spontaneous and chemically modulated activity and evoking neural responses. Overall, 3D HD-MEA demonstrated a high potential for the physiological investigation of neuronal circuit properties in 3D models.

## Materials and methods

### Preparation of acute cerebellar and brain slices

Animal maintenance and experimental procedures were performed according to the international guidelines of the European Union Directive 2010/63/EU on the ethical use of animals and were approved by the local ethical committee of the University of Pavia (Italy) and by the Italian Ministry of Health (protocol authorized following art.1, comma 4 of the D.Lgs. n. 26/2014 and approved on December 9th, 2017; authorization n. 1019/2023-PR).

Acute cerebellar and brain slices were obtained from C57BL6 mice (20–26 days old, either sex) following a standard procedure, as previously reported [8,49,50]. Briefly, mice were deeply anesthetized using halothane (Sigma-Aldrich) until paw and whisker reflexes completely disappeared, and euthanized by decapitation. Acute 220 µm parasagittal slices of the cerebellar vermis were cut using a vibroslicer (Leica VT1200S, Leica Microsystems). In parallel, acute 320 µm coronal brain slices containing the hippocampus were obtained on a second vibroslicer (Leica VT1200S). In some cases, acute 300 µm brain slices containing the prefrontal cortex were also obtained. During the whole procedure, slices were maintained in ice-cold Krebs solution containing (mM): 120 NaCl, 2 KCl, 1.2 $MgSO_4$, 26 $NaHCO_3$, 1.2 KH2PO$_4$, 2 $CaCl_2$, and 11 glucose, equilibrated with 95% $O_2$-5% $CO_2$ (pH 7.4). Slices were then recovered for at least 1h in Krebs solution, at room temperature. During recordings, slices were continuously perfused with Krebs solution (2 ml/min) using a peristaltic pump (Ismatec). When specified, 3 µM TTX (tetrodotoxin, Tocris) was added to the Krebs solution. When specified, for experiments with cortico-hippocampal slices, Krebs solution was modified to increase tissue excitability (increasing [K$^+$] to 8 mM, [Ca$^{2+}$] to 4 mM, and abolishing [Mg$^{2+}$]). For experiments on slices containing the prefrontal cortex, a modified ACSF (mACSF) with the following composition was also used when specified (mM): 124 NaCl, 3.5 KCl, 1 $MgCl_2$, 1.25 NaHPO$_4$, 1.2 $CaCl_2$, 26 $NaHCO_3$, and 10 glucose, equilibrated with 95% $O_2$-5% $CO_2$ (pH 7.4) [51]. Only for slices containing the prefrontal cortex the selective GABA-A receptor blocker SR95531 (10µM, gabazine, Abcam) was added to the mACSF when specified.

### Preparation of brain spheroids

The experimental procedures and animal care complied with the European Communities Parliament and Council Directive of 22 September 2010 (2010/63/EU) and with the Italian D.L. n. 26/2014, and were approved by the Italian Ministry

of Health (protocol number 75F11.N.6JI, 08/08/18). All possible efforts were made to minimize animal suffering and the number of animals used.

Rats (Sprague Dawley 200–250 g) were housed at the animal care facility of the Department of Pharmacy (DIFAR), University of Genova, Italy, at constant temperature (22 ± 1°C) and relative humidity (50%), under a light-dark schedule (lights on 7 AM–7 PM), and with free access to standard pellet diet and water. The development of brain spheroids (neurospheres) from primary embryonic neuron cultures was obtained following standard procedures as reported previously [52–54]. Briefly, rats were deeply anesthetized until paw and whisker reflexes completely disappeared, and euthanized by decapitation. Primary corteces of rat embryos E18-19 were isolated in HBSS without $Ca^{2+}$ and $Mg^{2+}$ and digested in a solution of 0.125% Trypsin + DNAse 50 µg/ml at 37°C for approximately 18–20 minutes. The proteolytic action of trypsin was blocked by the addition of 10% FBS medium (Neurobasal + B27 + Glutamax-100 1% + 10 ug/ml Gentamicin). The medium with 10% serum was removed and the tissue washed twice with fresh FBS-free medium. Then, by using a Pasteur pipette with a narrow tip the tissue was mechanically dissociated, counted in a haemocytometer to assess cell yield, and the cell suspension was diluted to the desired concentration. Spheroid structures were obtained by depositing the cells directly into 96-well plates (Biofloat) fabricated with ultra-low attachment plastic. It was possible to generate spheroids of different sizes by increasing the number of cells plated in each well. Typically, 12,000–13,000 cells per well were plated, which at the end of the development period gave rise to spheroids around 500 microns in diameter. Spheroid formation occurs quickly, and 48 hours after plating it was possible to appreciate their presence under the microscope. The plating medium was composed by Neurobasal + B27 + Glutamax-100 1% (ThermoFisher -Gibco) + 10 µg/ml Gentamicin, after 3 days in culture, 50% of the total volume of medium was replaced (to eliminate any debris) and finally, on day 5 *in vitro*, the medium was replaced 80% with BrainPhys + SM1 + 1% (StemCells Technologies) Glutamax-100 + 50 µg/ml Gentamicin. The culture of the spheroids was maintained in an incubator at 37°C, 5% $CO_2$ and 95% humidity for approximately 18–25 days with medium changes at 50% every 2–3 days. Then the individual spheroids were transferred directly onto the 3D high-density micro-electrode array (HD-MEA) chip for acute evaluation of electrical activity.

## Preparation of cortical and spinal cord organoids

Human induced pluripotent stem cells (hiPSCs) were generated from one healthy individual and cultured as previously described [55,56]. Briefly, hiPSCs were transferred from a 6 well to a 100 mm plate and cultured to 80–90% confluence. In the first day of preparation, the hiPSCs were dissociated with 5 ml of Accutase (VWR, Cat. 10761-312) per 100 mm plate and incubated 10 minutes at 37°C, in a 5% $CO_2$ incubator. The cells were then transferred on a 50 ml conical tube, centrifuged and resuspended in Essential 8 medium (E8, ThermoFisher, Cat. A1517001) supplemented with 10 µM Rock Inhibitor (RI, Tocris, Cat. 1254). Cells were counted to obtain 3 million cells per 1 ml of media; then 1 ml of suspension was added to each well of an Aggrewell 800 Plates (24-well, Stemcell, Cat. 34811) containing 1 ml of E8 (10µM Rock Inhibitor) for a total volume of 2 ml per well. Aggrewell were previously treated with 500 µl Anti-Adherence Rinsing Solution (AARS, Stemcell, Cat. 07010). The Aggrewell plate was then centrifuged at 200 g for 5 minutes to distribute cells in the microwell and then incubated 24 hours at 37°C, in a 5% $CO_2$ incubator to allow for aggregation. Next day, organoids were transferred to a 10 cm Ultra Low Attachment plate for the differentiation. Media for cortical organoids contained 10 ml of Neural Induction Media composed of Essential 6 (E6, ThermoFisher, Cat. A1516401) supplemented with 2.5 µM Dorsomorphin (DM, Sigma, Cat. P5499-25MG), 10 µM SB-431542 (SB, Selleck Chemicals, Cat. S1067), and 10 µM RI. From day 1 to day 5 the media was composed of E6 supplemented with 2.5 µM DM and 10 µM SB. From day 6 to day 25 the media was composed of NeuroBasal-A (NB-A, ThermoFisher, Cat. 12349015) NeuroBasal-A (NB-A, ThermoFisher, Cat. 12349015) supplemented with 20 ng/ml FGF2 (R&D Systems, Cat. 233-FB) and 20 ng/ml EGF (R&D Systems, Cat. 236-EG). From day 26 to day 44 the media was composed of NB-A supplemented with 20 ng/ml BDNF (PeproTech, Cat. 450-02) and 20 ng/ml NT3 (R&D Systems, Cat. 267-N3-005/CF). Media changes were performed daily until day 16, then once every 2 days until day 43 and every 3–4 days after that. After day 43 the media was composed only of NB-A.

Media for spinal cord organoids contained 10 ml of Neural Induction Media composed of E6 supplemented with 2.5 µM DM, 10 µM SB-431542 and 10 µM RI. During the following 42 days the media was supplemented with different compounds to address the differentiation towards the spinal cord neurons. From day 1 to day 2: E6 supplemented with 2.5 µM DM, 10 µM SB-431542. From day 3 to day 4: E6 supplemented with 2.5 µM DM, 10 µM SB-431542, 3 µM GSK3 inhibitor (CHIR, Selleck Chemicals, Cat. S1263). From day 5 to day 9: NB-A (+) supplemented with 10 ng/ml fibroblast growth factor 2 (FGF2, R&D Systems, Cat. 233-FB), 20 ng/ml endothelial growth media (EGF, R&D Systems, Cat. 236-EG), 0.1 µM retinoic Acid (RA, Sigma, Catalog R2625), 3 µM CHIR. From day 10 to day 17: NB-A (+) supplemented with 10 ng/ml FGF2, 20 ng/ml EGF, 0.1 µM RA, 3 µM CHIR, 0.1 µM smoothened agonist (SAG, Sigma Millipore, Cat. 566660). From day 18 to day 23: NB-A (+) supplemented with N-2 diluted 1/100 (ThermoFisher, Cat. 17502048), 20 ng/ml BDNF, 50 nM cAMP (Sigma, Cat. D0627), 200 nM L-acorbic acid (LAA, Fisher, CAT. 50-990-141), 10 ng/ml insulin-like growth factor-I (IGF, Preprotech, CAT. 100-11). Starting from day 24: NB-A (+) supplemented with N-2 diluted 1/100, 20 ng/ml BDNF, 50 nM cAMP, 200nM LAA, 10 ng/ml IGF. Media changes were performed daily until day 6, then once every 2 days until day 43 and then every 3–4 days.

## Viability fluorescence analysis

Cerebellar slices were loaded with calcein AM, a live cell-permeant dye, by incubation for 40 min in Krebs solution containing 20 µM calcein AM (acetoxymethyl ester of calcein; ThermoFisher Scientific). The slices were then washed three times in normal Krebs solution and fixed for at least 2 hours at RT or overnight at 4 °C in PBS PAF 4%. Slices were then washed 5 min in PBS for three times and mounted with Fluoroshield mounting medium with DAPI (Abcam, UK). The 220 µm thick cerebellar slices were mounted between two coverslips in order to analyze both sides; images were then acquired with TCS SP5 II LEICA (Leica Microsystem) equipped with an inverted microscope LEICA DM IRBE (UniPV PASS-BioMed facilities) and analyzed with ImageJ [57]. To estimate the percentage of vital granule cells, the ratio of cells loaded with calcein AM vs. granule cells stained only for DAPI (vital + nonvital cells) was manually calculated. The cell viability was analyzed on 20 fields per sample, on at least three different planes every two to avoid double counting.

## Voltage-sensitive dye imaging (VSDi)

VSDi recordings were performed on acute cerebellar slices, as in [49,58,59]. Briefly, slices for optical recordings were incubated for 30 minutes in oxygenated Krebs solution added to a 3% Di-4-ANEPPS (Invitrogen) stock solution and mixed with an equal volume of fetal bovine serum (Invitrogen), reaching the final dye concentration of 2 mM. The slices were then rinsed with Krebs solution and transferred to the mounting stage of an upright epifluorescence microscope (Slicescope, Scientifica Ltd). A proper set of filters (excitation filter λ = 535 ± 20 nm; dichroic mirror λ = 565 nm; absorption filter λ > 580 nm) was used to project the light to the slice and acquire the fluorescence signal, passing through a 20X water immersion objective (XLUMPlanFl, 0.95 numerical aperture; Olympus) and collected by a CCD camera (MICAM01 SciMedia, Brain Vision). The imaging system was connected to a PC unit through an input/output interface (Brain Vision), controlling illumination pattern, stimulation, and data acquisition. With this asset, the pixel size was 4.5 x 4.5 µm. Fluorescent signals were acquired using BrainVision software, with a sampling rate of 0.5 kHz. The mossy fiber bundle was stimulated with a tungsten bipolar electrode (WPI) connected to a stimulation unit through a stimulus isolator (stimulus intensity 15V, duration 250 µs). Single-pulse stimulation was delivered at 0.1 Hz, and every VSDi trace was obtained averaging ten repetitions, to improve signal-to-noise ratio [49]. Data analysis was performed as previously described [49]. Firstly, traces were filtered (3 x 3) using a cubic and a spatial filter. Secondly, ad hoc routines in Matlab (Mathworks) were implemented to analyze the responses to stimulation, as rapid and transient increases in fluorescence exceeding the baseline average ΔF/F by more than 2.5 standard deviations. This procedure allowed maps of the spatial distribution of granular layer responses to stimulation to be obtained. The percentage of the granular layer activated by the stimulus was calculated by dividing the number of pixels showing a response to the stimulation by the total number of pixels sampling the granular

layer and multiplying by 100. Statistical comparison was performed using unpaired Student's *t* test, with significance level < 0.05. Data are reported as mean ± SEM (standard error of the mean).

## Characterization of electronic chip performance

To evaluate the circuit performance of the chips, we calculated the signal-to-noise ratio (SNR) of the on-chip amplifiers. This is defined as the ratio of signal power (Psignal) to noise power (Pnoise) for each electrode:

$$SNRe = \frac{Psignal}{Pnoise}$$

The SNR for an individual chip (SNRc) was then calculated as the median value. A total of N = 5 planar and N = 5 3D HD-MEAs were used in this analysis. The injected signal was a sinusoidal wave with a peak-to-peak amplitude of 100 µV and a frequency of 1 kHz. Prior to injection, the signal characteristics were verified using an oscilloscope (Siglent SDS2104X Plus).

It is important to note that this evaluation was performed under dry conditions, specifically to isolate and compare the intrinsic circuit performance of the planar and 3D HD-MEAs. As such, the results do not reflect the chips' actual ability to detect voltage fluctuations from background noise in operational (wet) conditions, which is better illustrated in Figs 3 and 4. Instead, these measurements demonstrate that the circuit performance of both chips is comparable, and that the enhanced signal detection observed with the 3D HD-MEA is attributable to its electrode architecture rather than differences in amplifier performance.

## Electrophysiological recordings

Electrophysiological recordings on cerebellar and cortico-hippocampal brain slices were performed using planar and 3D HD-MEA from 3Brain AG (CorePlate 1W 38/60 and CorePlate 1W-3D 38/60/90). Both planar and µneedles 3D HD-MEAs rely on the same electronic chip. In particular, the chip provides 4096 electrodes arranged in a 64x64 grid with an electrode pitch of 60 µm (total recording area 3.8 x 3.8 mm²). Each electrode integrates an amplification stage and a multiplexing architecture as presented in [60], enabling simultaneous recordings from all the 4096 electrodes at a sampling rate of 20kHz. Differently from the original architecture [60], a switching circuitry allows to route each individual electrode to a current signal generator integrated in the acquisition system (BioCAM Duplex from 3Brain AG), enabling electrical stimulation from the HD-MEA chip. Planar electrodes consist of a square of 21 µm side covered by a thin layer of platinum, while the 3D µneedle electrodes provide a gold pedestal of 30 µm width and 15 µm height. On top of the pedestals, two versions of the µneedles were realized: a large one of 26 µm width and 90 µm height (Fig 1A) and a thinner one of 14 µm width and 65 µm height (Fig 1B). The electrodes are insulated by a layer of parylene leaving an electrochemically accessible electrode site on the tip (Fig 1C). In "*Improved viability and functional network preservation*" chips with different pedestal sizes were used (44, 36, 30 µm).

In cortico-hippocampal slices, a 10 Hz high-pass filter was applied to remove slow oscillations and only the electrodes in the cortical region with a mean firing rate higher than 0.1 Hz were considered for the analysis. In cerebellar slices, a 50 Hz high-pass filter was applied during signal acquisition and only the electrodes with a mean firing rate higher than 0.5 Hz were considered for the analysis.

To perform a direct comparison of the efficiency in recording signals of the µneedle electrodes vs. planar ones, each slice was measured on both a planar and a 3D HD-MEA. A total of 6 cerebellum slices and 4 cortico-hippocampal slices were tested from 3 different animals. On the planar HD-MEA, the slice was stabilized using a platinum anchor with wires of 100 µm width, while no holder was used on the 3D HD-MEA. Recordings consisted of 3–5 minutes of spontaneous activity performed 5 minutes after the slice was positioned on the chip. To test whether 5 minutes were sufficient to

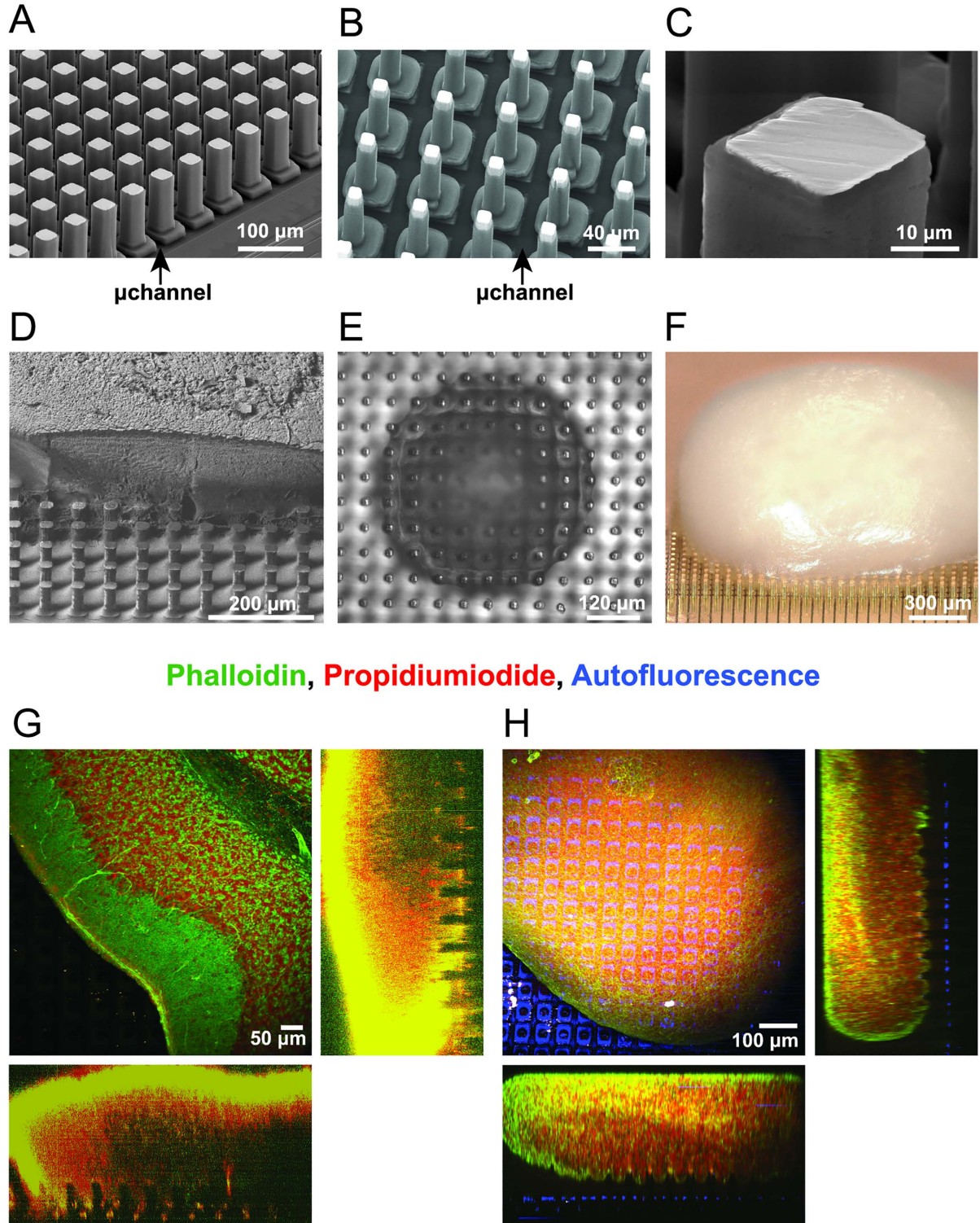

**Phalloidin, Propidiumiodide, Autofluorescence**

**Fig 1. 3D HD-MEA chip characterization and tissue penetration.** A, B) SEM images of a 3D HD–MEA with larger (a) or smaller (b) μneedles. Arrows indicate microchannels (μchannel). C) Detailed view on the sensing area of a single μneedle. D) A 250μm thick cerebellar slice fixed on the 3D HD-MEA and used for SEM. A longitudinal cut with a Focused Ion Beam shows the μneedles penetrating the tissue without bending or deforming it. E)

Phase-contrast image of a brain spheroid mounted on a 3D HD-MEA showing the intimate contact of the tissue with the µneedles. Notice that the image is focused on the plane of the µneedle tips. F) Digital microscope image (100x magnification) of a cortical brain organoid positioned on the 3D HD-MEA. G, H) Two-photon microscopy volume scans of a brain slice (G) and a brain-spheroid (H) stained with phalloidin (*green*) and propidium iodide (*red*), while chip autofluorescence appears in *blue*. Note the penetration of µneedles into the tissue.

achieve a stable coupling for both planar and 3D chip, we measured the number of channels detecting spiking activity for the first 25 minutes after a cerebellar slice was placed on the chip. Spike detection was performed using Brainwave 5 with a hard threshold at −100 µV and considering only channels showing a spike rate higher than 10 Hz. While during the first few minutes the planar chip showed a slight increase in the number of active channels ($3.9 \pm 2.5\%$, N = 5), the 3D chip showed a slight decrease in the same parameter ($8.1 \pm 2.9\%$, N = 5; S1A Fig). However, the number of active channels remained stable from 5 to 25 minutes after the slice positioning for both chips, as evident in S1B Fig, where the data are normalized on the number of active channels at 5 minutes (planar vs. 3D unpaired Student's t test p = 0.38).

TTX modulation of spontaneous activity in cerebellar slices was recorded by acquiring activity for about 4 minutes in standard Krebs solution, followed by perfusion of Krebs with 3 µM TTX. Each slice was monitored for the following 3 minutes or until the activity completely disappeared. 10 slices from 3 different animals were tested for both planar and 3D HD-MEA.

On a subset of cerebellar slices mounted on the 3D HD-MEA, electrical stimulation was also performed by releasing a biphasic stimulation from two adjacent electrodes selected in correspondence with the mossy fibers in the granular layer of a lobule. Stimulation intensity was set at 80 µA for 120 µs for the first phase followed after 25 µs by a second phase with −40 µA delivered for 50 µs. Different lobules of the same slice were tested. The amplifiers were blanked for 500 µs after each pulse to reduce the artifacts produced by the electrical stimulation. The nature of the response was assessed using 10 µM NBQX (selective antagonist of AMPA receptors; Abcam) and 3µM TTX (Abcam).

In prefrontal cortex slices, a 100 Hz high-pass filter was applied to remove slow oscillation and only the channels with a mean firing rate above 0.1 Hz were considered for the analysis.

Spontaneous activity of cortical brain spheroids (n = 6) was measured at DIV 19–20 by using the 3D HD-MEAs. Recordings consisted of acquiring a few minutes of basal activity in acute conditions: three spheroids per chip were transferred with a pipette with a large tip on the 3D HD-MEA previously treated with a plasma cleaner process to make the surface hydrophilic. No anchor was used, and the spheroids started showing spontaneous spiking activity immediately after the placement on the chip. Two different sizes of brain spheroids (about 400 µm and 600 µm) were tested. During the recording, the spheroids were kept in a culture medium with a 5% $CO_2$ humidified flux and at approximately 35°C.

Cortical and spinal cord organoids were measured respectively at DIV 219 and DIV 131 by using the 3D HD-MEA. The organoids were transferred on the chip using a plastic Pasteur pipette with a large tip, and a 3D-printed silicon holder was placed on the chip well to keep the organoid in the center of the electrode array and facilitate the µneedles penetration. Spontaneous activity was detected right after the organoid placement over the chip, and an initial 2-minute baseline recording was performed. Then, 1mM KCl was added to the organoid medium, and the activity was recorded for another 2 minutes. Finally, KCl concentration was increased to another 5mM, and the electrical activity was recorded again for 2 minutes. During the whole recording, the organoid was kept in a culture medium with a 5% CO2 humidified flux at approximately 37°C.

### Electrophysiological recordings: Data analysis

Data were acquired, visualized, and stored using version 4 and 5 of BrainWave software from 3Brain AG. The software allows superimposing the slice image positioned on the chip with the activity map for anatomically matching the recordings with the brain regions (e.g., Fig 3A and 3B). Spikes were extracted from cerebellar slice recordings using the algorithm presented in Multhmann et al., 2015 [61] and, for cortical hippocampal slices, the precise timing spike detection (PTSD) algorithm as described in Maccione et al., 2009 [62]. Detected events in cerebellar slices were then sorted as

described in [63]. In cortical hippocampal slices, the sorting tool of BrainWave 5 software was used. Specifically, spike waveforms were analyzed using principal component analysis for feature extraction, followed by K-means clustering with Gap Statistics to determine the optimal number of clusters. A maximum of four clusters (i.e., four cells) could be sorted per electrode. An active unit was considered for the analysis if the spike rate was higher than 0.5 Hz in cerebellar slices and 0.1 Hz in cortico-hippocampal brain slices and if its position was within the considered brain region. For the "*Increased recording capabilities in detecting spiking activity in acute slices*" and "*Accelerated effect of drugs modulating electrical activity*" data analysis, a total number of 6 and 20 cerebellar slices (from 3 animals) were used. All datasets were initially assessed to ensure the distribution followed a Gaussian curve using the Shapiro–Wilk normality test. The group comparison of the "*Increased recording capabilities in detecting spiking activity in acute slices*" data analysis was performed using an unpaired Student's *t* test. The "*Accelerated effect of drugs modulating electrical activity*" data analysis was conducted using a two-way ANOVA followed by Tukey's multiple comparison post-hoc test. Statistical significance was set at $p < 0.05$ and all the statistical analyses were performed using GraphPad Prism (9.0, GraphPad Software, La Jolla, CA).

For prefrontal cortex slices (7 slices from 4 animals), the activity correlation between different units in the prelimbic area was calculated using BrainWave 5 software from 3Brain AG and custom Python scripts. The correlation analysis provides the total number of direct links between units (supposed connections) [64,65]. This estimate was divided by the total number of selected units to normalize the data for each slice (correlation index, CI). The firing parameters (basal frequency, CV, and CV2) for each unit and the response to stimulation (LFPs, raster and PSTH plots) were analyzed using NEURO-Pulse [66]. Both correlation and basal activity data were processed using GraphPad Prism (9.0, GraphPad Software, La Jolla, CA). Statistical analysis was performed using paired Student's *t* test unless otherwise stated, and significance was set at $p < 0.05$.

For brain spheroid experiments, spike detection was performed using the PTSD algorithm as indicated above, while spike sorting on individual electrodes was computed using a principal component analysis (PCA) with 3 features and a standard k-means clustering algorithm. A unit was considered active if the spike rate was higher than 0.1 Hz and was located below the electrode area covered by the spheroids. In total, 6 spheroids of two different sizes were measured from a single preparation.

All the recordings made with the cortical and spinal cord organoids were analyzed by performing a spike detection using the PTSD algorithm (see above). Each unit with a spike rate higher than 0.5 and 0.1 Hz, respectively, was considered active; each series of at least 5 consecutive spikes with 100 ms inter-spike interval was considered a burst. Cross-correlation among couples of active units was computed on the spike sequences within a 30 ms correlation window, binned in intervals of 3 ms. The output of the cross-correlation computation was normalized to obtain a "correlation value" between 0 and 1 [64,65]. Statistical analysis was performed by considering all the active units for each phase of recording. Differently, the statistical analysis of the correlation value was conducted by randomly sampling 100 values for each phase of recording. One-way ANOVA followed by Tukey post-hoc test was used, statistical significance was set at $p < 0.05$, and all statistical analyses were performed using GraphPad Prism (9.0, GraphPad Software, La Jolla, CA).

## Two-photon microscopy

Cerebellar slice and brain spheroids were incubated in 0.2% Triton X-100 (Acros Organics) for 20 min and then transferred in a staining solution containing 0.1% Triton X-100, 200 nM Phalloidin-Atto488 (Atto-tech) and 2.5 µg/ml propidium iodide (Sigma) and allowed to incubate for 1 hour. The staining solution was removed and followed by 3 washing steps with PBS. Finally, the samples were mounted in 70% glycerol on top of a 3D HD MEA chip glued in the middle of the cavity of a 35 mm cell culture dish (ibidi). The volume scan was performed using an upright MPX multiphoton microscope (Prospective Instruments) with a 20x Olympus water dipping Objective (NA 1.0, WD 2.0 mm). Three channels were simultaneously recorded to capture the 2P autofluorescence of the chips, the actin fibers, and the cell nuclei. The step size for the 3D volume scans was set to 2 µm. Three-color images were merged using ImageJ (Version 1.52v).

## Results

### 3D µneedles structure and tissue penetration

The dense structure of µneedles fabricated on top of the 3D HD-MEA is visible in the scanning electron microscope (SEM) images in Fig 1. Two different sets of µneedles were used for recordings in brain slices (µneedle height 90 µm, width 26 µm, Fig 1A) and organoids (µneedle height 65 µm, width 14 µm, Fig 1B). The pedestals at the base of the µneedles form a microchannels grid, as indicated by the arrow in Fig 1A,B. The µneedles are characterized by an internal bulk in gold, surrounded by a thin insulating layer of parylene, as shown in Fig 1C. The most crucial property of a 3D µneedle is its ability to penetrate the tissue. To test this property in the two sets of µneedles, we fixed cerebellar slices or spheroids over the 3D chip (see Methods for details) and used scanning electron microscopy (SEM), differential interference contrast (DIC) microscopy, or two-photon microscopy scanning for assessment. In particular, the tissue of the cerebellar slice was longitudinally cut using a Focused Ion Beam procedure, leaving a portion of the internal layers exposed. The µneedles penetrated up to a third of the total thickness of the tissue, which remained in contact with the base of the chip, as evident in the SEM image in Fig 1D. The set of thinner µneedles in Fig 1B, developed for smaller biological models such as organoids or spheroids, was tested for penetrating capabilities using spheroids. DIC microscopy revealed that the thinner µneedles penetrated for all their length into the tissue (Fig 1E). The tissue engulfs the µneedles without losing its spherical structure or showing evident signs of debris or cell detachment. An entire stack of images from the base of the chip to the top of the spheroid is shown in S1 Movie. Penetration without provoking evident damage in organoids is also shown in Fig 1F. Tissue penetration was also assessed using two-photon microscopy scanning for both cerebellar slices (Fig 1G) and spheroids (Fig 1H).

### Viability and functional network preservation

To test whether the microchannel arrangement of 3D HD-MEA can improve tissue viability, we performed vitality and functionality tests on three different microchannel configurations (see Materials & Methods). Cerebellar slices are an ideal model to test vitality since the high density of granule cells in the cerebellar cortex [67,68] makes them more susceptible to hypoxia. Acute cerebellar slices were placed on chips with different microchannel widths, modifying the area in direct contact with the slice. Reducing the area of direct contact with the slice should improve tissue vitality, increasing the surface available for exchanges with the perfusing solution circulating in the microchannels. To test this hypothesis, three conditions were implemented using chips characterized by: small microchannels (SMC, with 16 µm channels width and 44 µm pedestal width, corresponding to a contact area on all the chip of $7.9\,mm^2$); intermediate microchannels (IMC, with 24 µm channels width, and 36 µm pedestal width, corresponding to a contact area on all the chip of $5.3\,mm^2$); large microchannels (LMC, with 30 µm channels width, and 30 µm pedestal width, corresponding to a contact area on all the chip of $3.6\,mm^2$). The vitality of slices on these chips was compared to those positioned on a planar chip (2D). After one hour of continuous perfusion with oxygenated Krebs solution, the slices were gently removed from the chips or coverslips and loaded with 20 µM calcein AM, a compound that becomes fluorescent when hydrolyzed by esterases in living cells [69]. Confocal microscopy images were obtained from both sides of each slice, allowing calcein staining quantification (Fig 2A). The number of living cells on the lower side (in contact with the chip or coverslip) was compared to that on the free surface on the upper side (see Methods). The vitality of the lower side, compared to that on the upper side, was $38\pm4\%$ in the slices mounted on coverslips and $76\pm3\%$ in those placed on LMC 3D HD-MEA chips (Fig 2A). This difference matches that in the contact area of the lower side, which was $3.6\,mm^2$ on 3D chips and $15\,mm^2$ on planar chips (Fig 2B). These results demonstrate that microchannels enhance tissue viability, presumably by improving oxygenation and nutrient diffusion in the lower layers of the slice.

Enhanced tissue viability may not prevent perturbation of network activity by thousands of µneedles, which might alter cell functioning or disrupt local connectivity. To address this issue, we performed VSDi to track neuronal membrane depolarization with good spatial resolution [49]. Cerebellar slices were placed on the 3D HD-MEAs and the response to

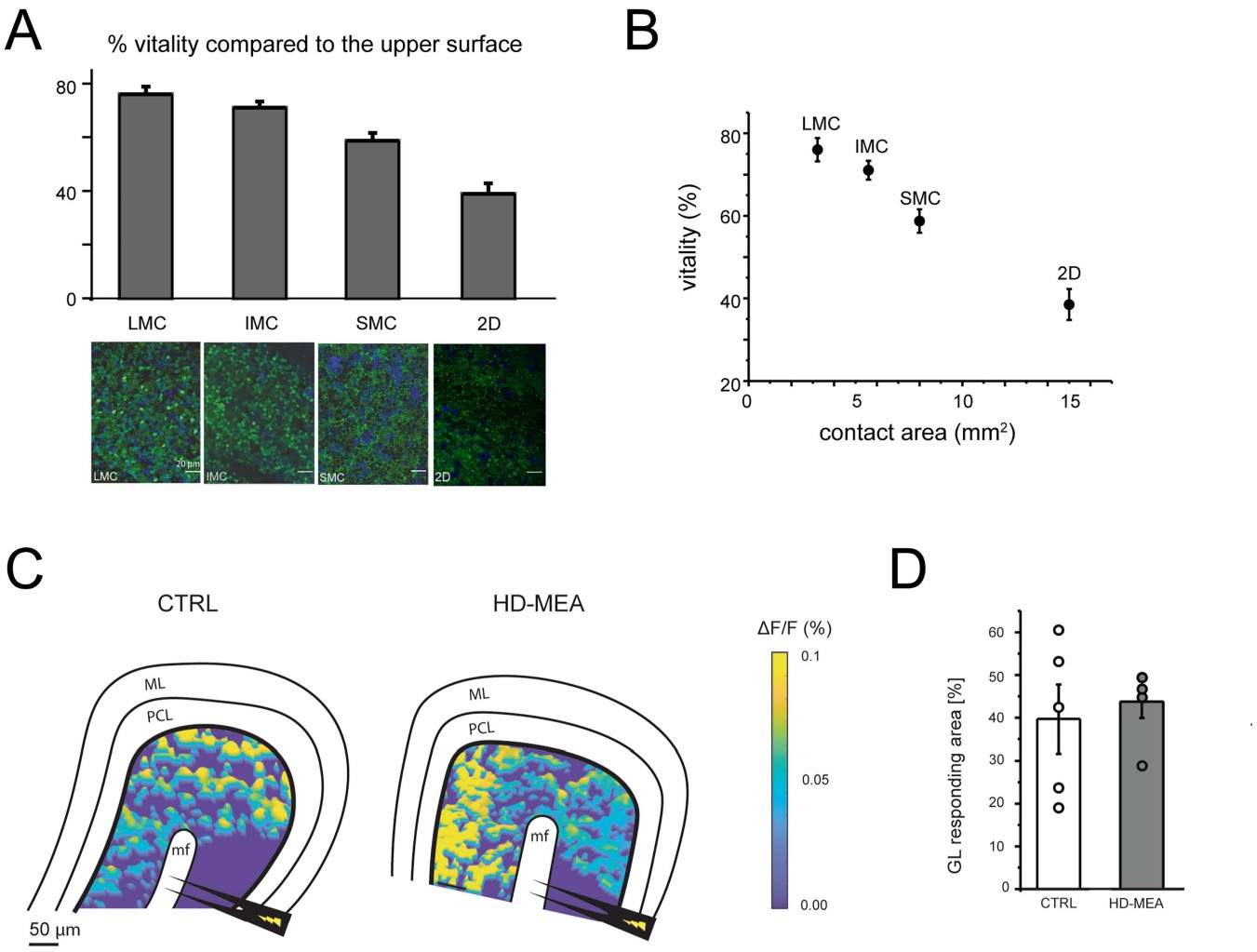

**Fig 2. Tissue viability and network preservation in cerebellar slices on planar and 3D HD-MEA.** A) The histogram shows the vitality in the lower surface (in contact with the chip) compared to the upper surface (in contact with the medium) of slices positioned on the planar (2D) or 3D HD–MEA chip for 1h before calcein staining. Representative confocal images at 40x magnification of cerebellar slices loaded with calcein AM in the LMC, IMC, SMC, and 2D are shown below. Living cells are stained in green; cell nuclei are stained in blue (DAPI). Scale bar = 20 μm. Note the higher percentage of living cells on 3D HD–MEA chips. B) The plot shows the percentage of viability of the lower surface compared to the upper surface for each configuration shown in A. The contact area for SMC, IMC, and LMC is estimated from the microchannel width. C) Example of pseudocolor maps obtained with VSDi recordings showing the spatial distribution of granular layer responses to mossy fiber stimulation in CTRL (*left*) and in a slice on the 3D HD–MEA (*right*). The stimulating electrode (*black*) is positioned over the mossy fiber bundle (MF). GL, granular layer. PCL, Purkinje cell layer. ML, Molecular layer. D) The histogram shows the percentage of granular layer area responding to mossy fiber stimulation in VSDi experiments. The area in the two conditions tested is not significantly different (data are reported as mean± MSE. As in the text, LMC, IMC, SMC, stand for 3D HD–MEA chips with large, intermediate, and small microchannels, while CTRL means placing the tissue on a coverslip.

mossy fiber stimulation was measured using VSDi. Again, the cerebellar granular layer is an ideal model for detecting a possible alteration in the spatial distribution of activity since granule cells are more densely packed than in other brain regions. Slices were placed either over the chip or on the bottom of the recording chamber (as in [49]). The percentage of the granular layer area responding to stimulation was not significantly different in the slices placed on the chip compared to controls (43.7±3.8% n = 5, 39.7±8.1% n = 5 respectively; p = 0.66, Fig 2C,D). No evidence of alterations in the granular layer activity caused by the μneedles was evident in our results.

## Recording capabilities in detecting spiking activity in acute slices

To assess the improved capabilities of the µneedle electrodes in detecting spiking activity, we first assessed the electronic chip performance of both planar and 3D HD-MEAs calculating the signal-to-noise ratio (S/N) in dry testing conditions. S2 Fig shows no significant differences, confirming that the amplification performance is consistent regardless of HD-MEA chip type and that the improved sensing capabilities shown in the following sections can be attributed to the µneedles fabricated on top of the electrodes.

Cerebellar and cortico-hippocampal slices were used to characterize spontaneous activity on HD-MEA chips. The slices were first placed on a planar HD-MEA chip and then transferred onto a 3D HD-MEA chip (or vice versa) and recorded for 3 minutes in each case. The 3D HD-MEA chip showed increased performance in detecting activity from the cerebellar Purkinje cell layer compared to the planar HD-MEA, as evident in the firing rate maps in Fig 3A. The activity map of the Purkinje cell layer recorded using the 3D HD-MEA matched the anatomical structure of the cerebellar slice with a finer grain compared to the planar HD-MEA chips (Fig 3A). In cerebellar slices, the recordings performed on the 3D chip showed a significantly larger number of active electrodes (80±19%, 390±49 on 2D vs 790±62 on 3D; n=6 slices, p=0.0006), number of sorted cells (102±23%, 371±45 on 2D vs 668±39 on 3D; n=6 slices, p=0.0005), and cells to electrodes ratio (12±5%, 1.05±0.01 on 2D vs 1.18±0.03 on 3D; n=6 slices, p=0.0126), compared to the same slices on the planar HD-MEA chip (Fig 3B). Moreover, following spike detection and sorting analysis, the active units in cerebellar slices over the 3D chip showed a significantly higher mean firing rate compared to those recorded using the planar HD-MEA (105±30%; 25±1.8 Hz on the 2D vs 51±5.4 Hz on the 3D; p=0.0010), together with a significant improvement of the peak-to-peak amplitude (46±14%; 76±5 µV on the 2D vs 111±9 µV on the 3D; p=0.0060;) (Fig 3C).

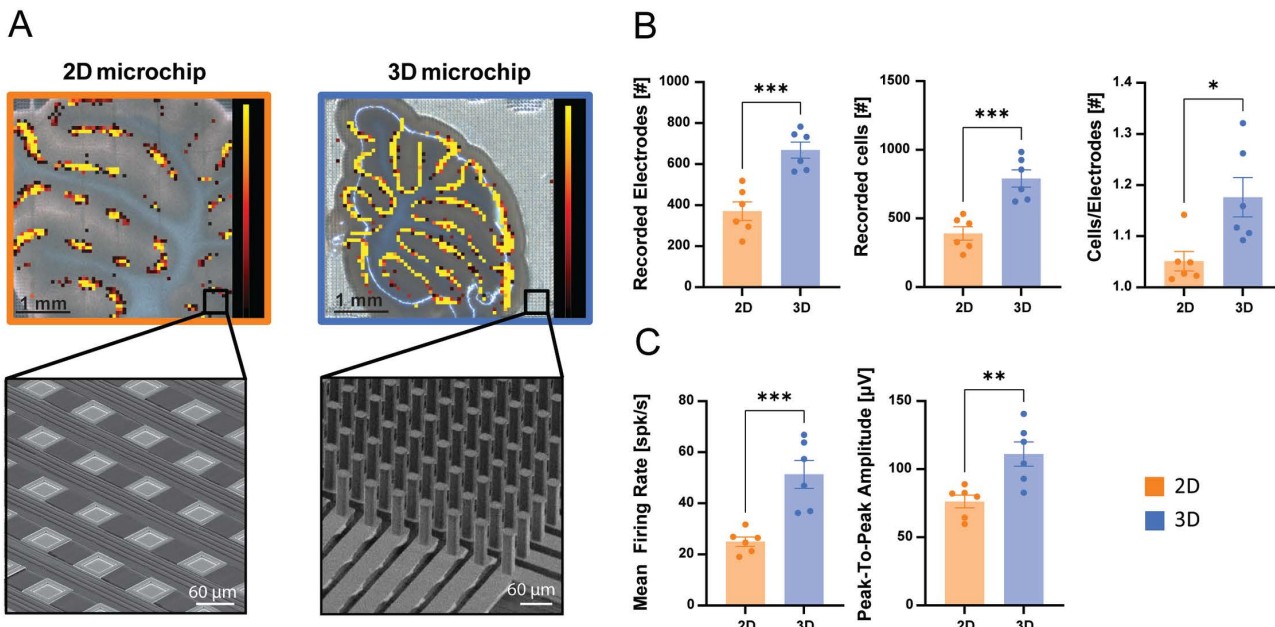

**Fig 3. Recording capabilities of 3D HD-MEA compared to planar HD-MEA in cerebellar slices.** A) Images of cerebellar slices overlaid with the corresponding electrical activity maps showing the mean firing rate detected by the HD-MEA electrodes (color scale bar: 0- 10 spikes/s; mean firing rate > 0.5 spikes/s). Left: slice placed on a planar HD-MEA chip; right: slice placed on a 3D HD-MEA chip. The insets show SEM images of the planar and 3D HD-MEA chips. B) The histograms show the number of electrodes recording at least one extracellular unit, the number of recoded cells, and the cells/electrode ratio for the planar and 3D HD-MEA chips. C) The histograms show the mean firing rate and peak-to-peak amplitude of the units recorded by the planar and 3D HD-MEA chips. In B and C, histograms report mean±MSE and asterisks indicate significant differences at * p<0.05, ** p<0.01, and *** p<0.001.

Similarly, the activity in the number of detected spikes maps from cortico-hippocampal slices was more evident using the 3D HD-MEA compared to planar HD-MEA (Fig 4A). The 3D HD-MEA showed a significantly larger number of active electrodes in the cortical area compared to the planar HD-MEA (92±85%, 56±17 on the 2D vs 154±25 on the 3D; n=3 slices, p=0.0384) (Fig 4C). Although the increase is not statistically significant, both the number of cells (Fig 4 B,C) and the number of detected spikes (Fig 4C) are higher in slices placed on the 3D HD-MEA compared to those on planar HD-MEA chips. Notably, the peak-to-peak amplitude of the detected spikes is significantly greater in the 3D HD-MEA than in the planar configuration (105%±30%, 166±7 on the 2D vs 257±30 on the 3D; n=3 slices, p=0.0221) (Fig 4C). Considering that cortical activity is characterized by a "random" and sparse firing pattern, it is likely that the µneedles could access inner layers where neurons are better preserved, increasing the number of active units that could be detected. In aggregate, these data show that the µneedles of the 3D HD-MEA significantly improved the recording capabilities of the electrodes in the tested brain regions.

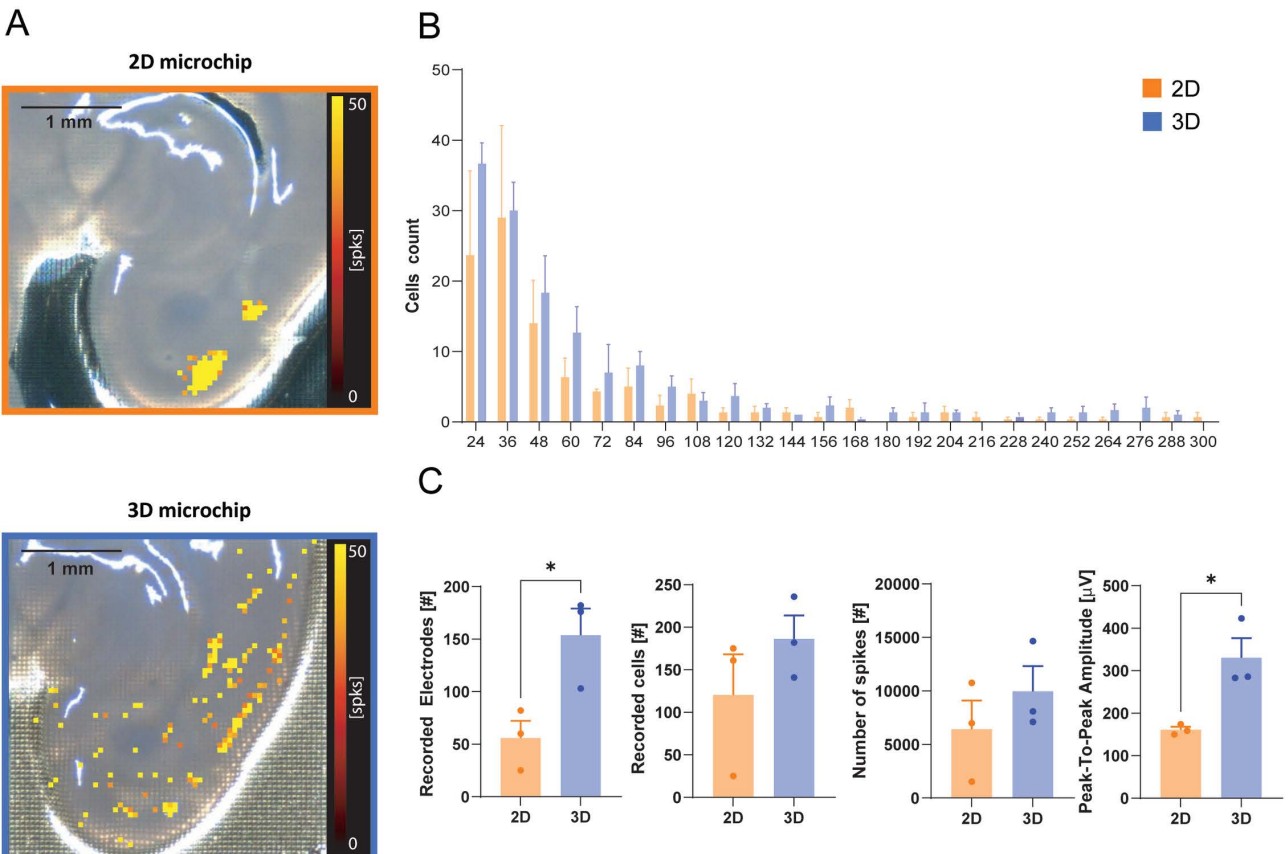

**Fig 4. Recording capabilities of 3D HD-MEA with respect to planar HD-MEA in cortico-hippocampal slices.** A) Images of cortico-hippocampal slices overlayed with the corresponding electrical activity maps showing the number of spikes detected in 3 minutes by the HD-MEA electrodes (color scale bar: 0-50 spikes). Top: slice placed on a planar HD-MEA chip; bottom: slice placed on a 3D HD-MEA chip. B) Distribution of the number of spikes by cell count in the planar and 3D HD-MEA chip. C) The histograms show the average number of active electrodes, average number of active cells, average total number of spikes and the peak-to-peak amplitude of the spikes recorded by the planar and 3D HD-MEA chip. In B and C, histograms report mean±MSE and asterisks indicate significant differences at * p<0.05.

## Accelerated effect of drugs modulating electrical activity

The µneedle pedestals avoid the tissue touching the bottom of the 3D HD-MEA chip, thus originating a grid of microchannels below the tissue itself (Fig 1G–H). Besides improving tissue viability, the solution flowing below the bottom surface is also expected to be improved, facilitating compound diffusion and therefore reducing the time of action. To test this possibility, spontaneous activity was recorded from cerebellar slices for one minute before and during perfusion of a Krebs solution containing 3 µM tetrodotoxin (TTX). As expected, spiking activity was progressively suppressed by TTX perfusion (Fig 5A) both in slices placed over a planar and a 3D HD-MEA. However, slices on the 3D HD-MEA showed a faster decrease

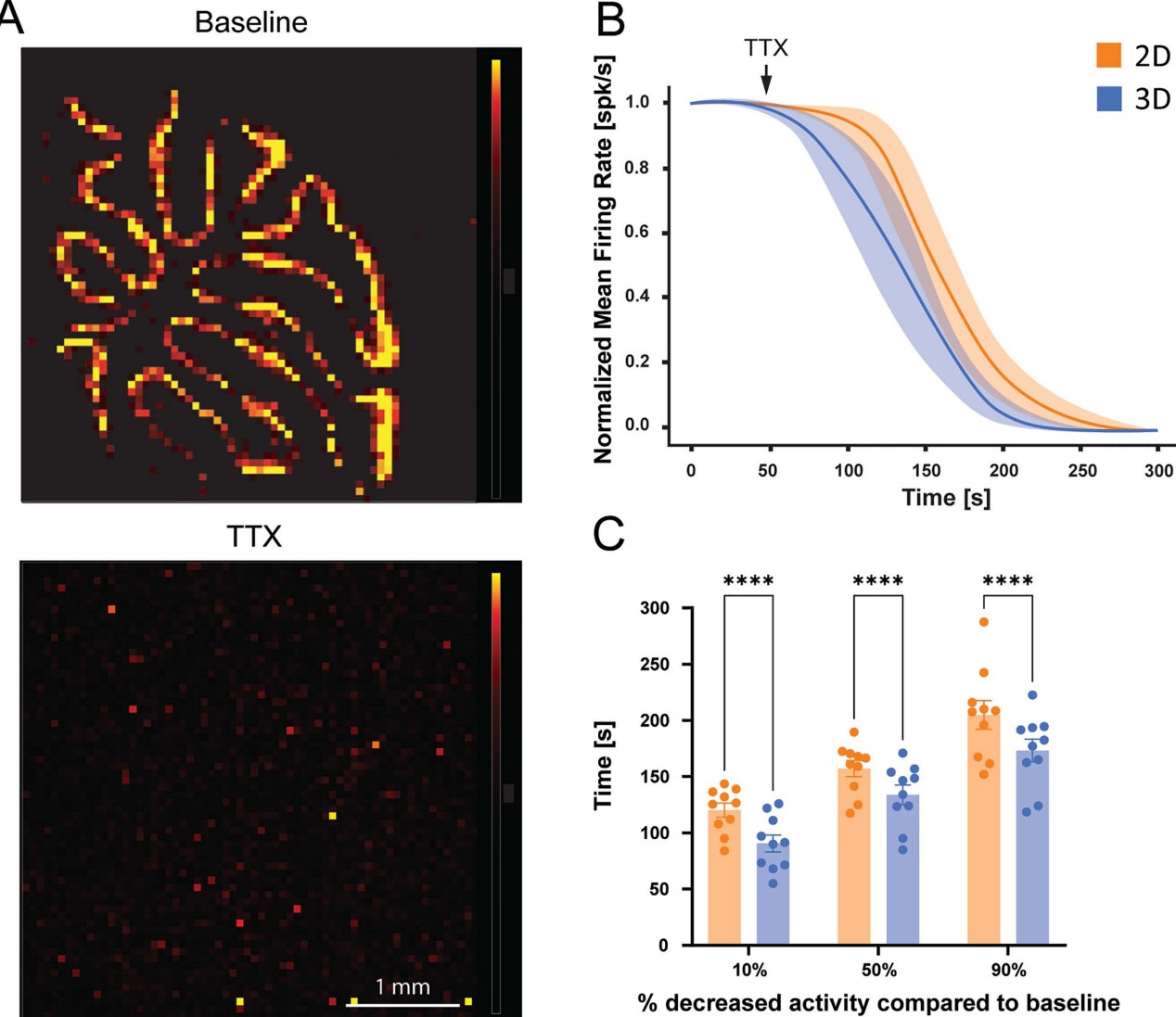

**Fig 5. Accelerated drug action with 3D HD-MEA compared to planar HD-MEA.** To test the effectiveness of drug perfusion, TTX was applied to acute cerebellar slices. A) Examples of electrical activity maps showing the distribution of active electrodes before (*top*) and after TTX application (*bottom*) in a slice placed on a 3D chip (color scale bar: 0-300 µV). B) The plot shows the normalized time course of the mean firing rate (MFR) of the active units following TTX perfusion over the cerebellar slices placed on planar and 3D HD-MEAs. The mean firing rate was normalized to 1 min basal activity recorded before TTX application. C) The histograms show the time needed to reach the 10%, 50%, or 90% decrease in firing frequency after TTX application in both conditions. Histograms report mean±MSE and asterisks indicate significant differences at * p<0.05, ** p<0.01, *** p<0.001.

in the mean firing rate after TTX application (at the 10%, 50%, and 90% of decrease; $p < 0.0001$ in all conditions, $n = 10$ slices), attaining full suppression of neuronal activity earlier than slices on planar HD-MEA (Fig 5B and 5C, S2 Movie). These results show that the action of active compounds is significantly accelerated using 3D HD-MEA.

## Stimulation capabilities

The planar and 3D HD-MEA chips are endowed with bidirectional electrodes that can be used to record and deliver electrical stimulation. In cerebellar slices, the granular layer response to mossy fiber stimulation is millisecond-fast [8,13], and therefore challenging to detect. Moreover, efficient stimulation requires tissue penetration to access inner mossy fibers. For these reasons, µneedles are expected to provide a crucial advantage for recording and stimulating evoked electrical responses. Our results show that mossy fiber stimulation evoked a local field potential (LFP) which propagated in each stimulated lobule (Fig 6A) with an average speed of $311 \pm 39$ mm/s ($n = 8$). While a fast response was consistently observed with 3D HD-MEA (Fig 6A and S3 Movie), the same protocol failed to activate the granular layer in planar HD-MEA recordings. Moreover, we tested the pharmacological manipulations of the granular layer response, which is known to be abolished by blocking AMPA receptors (10 µM NBQX, Fig 6A, S3 Fig) and voltage-dependent Na$^+$ channels (3 µM TTX, Fig 6A). The stimulation efficiently allowed signal propagation through the circuit, as evident from the responses detected in the Purkinje cells of the stimulated lobules (Fig 6 B,C). Purkinje cells receive synaptic contacts from granule cell axons, showing that the stimulation was effective in activating the network. Purkinje cell responses were also abolished by NBQX perfusion (Fig 6 B,C).

## Testing the 3D HD-MEA recording efficiency on challenging samples

Given the increased recording capabilities shown by the 3D chip in the previous tests, we challenged the device ability to record from demanding preparations (i.e., conditions where assessing the electrophysiological properties of the functional network is classically considered problematic). In particular, we recorded spontaneous activity in brain slices containing the prefrontal cortex (PFC), brain spheroids, and organoids.

## Spontaneous network activity of the prefrontal cortex in acute slices

Recently, much effort has been devoted to characterizing the prefrontal cortex (PFC) circuit. This should not be surprising, considering the high-level functions associated with this brain area. However, a detailed characterization of PFC neurons coordinated activity within the network is still lacking. This is due to the inherently difficult conditions for recording PFC spontaneous activity, mostly relying on manipulating neuronal excitability (see Discussion for more details). To test whether the increased recording capabilities of the 3D chip could help with this issue, we first recorded spontaneous activity in the prelimbic region (PrL) of the PFC in acute slices in physiological conditions (i.e., without increasing neuronal excitability with modified extracellular solutions). We recorded 983 channels showing basal spontaneous activity (> 0.1 Hz) in 7 slices, accounting for 43% of the 2284 electrodes sampling the PrL. The mean firing rate of these units was $0.71 \pm 0.09$ Hz. To further characterize the PrL activity, we modified neuronal excitability by perfusing first a modified ACSF (mACSF) with no Mg$^{2+}$ and increased K$^+$ concentration (see Methods) and then adding an inhibitory transmission blocker (10 µM gabazine) (Fig 7A and S4 Fig). As expected, both these modifications increased network excitability. Interestingly, we also observed the generation of spontaneous large events traveling along the cortex as LFP with superimposed spike bursts (Fig 7B and S4 Movie). This latter phenomenon is inherently a network event and was evident in a subset of experiments. Therefore, we characterized the two groups of slices (with or without propagating LFPs) to detect possible differences in basal activity that could explain the insurgence of paroxysmal events after gabazine perfusion only in a subset of slices. No difference in the basal firing properties was detectable in the group of slices devoid of LFP ($n = 3$; indicated from now on as Group 1) and in the group of slices showing LFPs ($n = 4$; Group 2) (S5 Fig). In physiological conditions, the firing frequency (Group 1: 363 units, $0.63 \pm 0.07$ Hz; Group 2: 620 units, $0.79 \pm 0.12$ Hz; unpaired Student's t test $p = 0.25$)

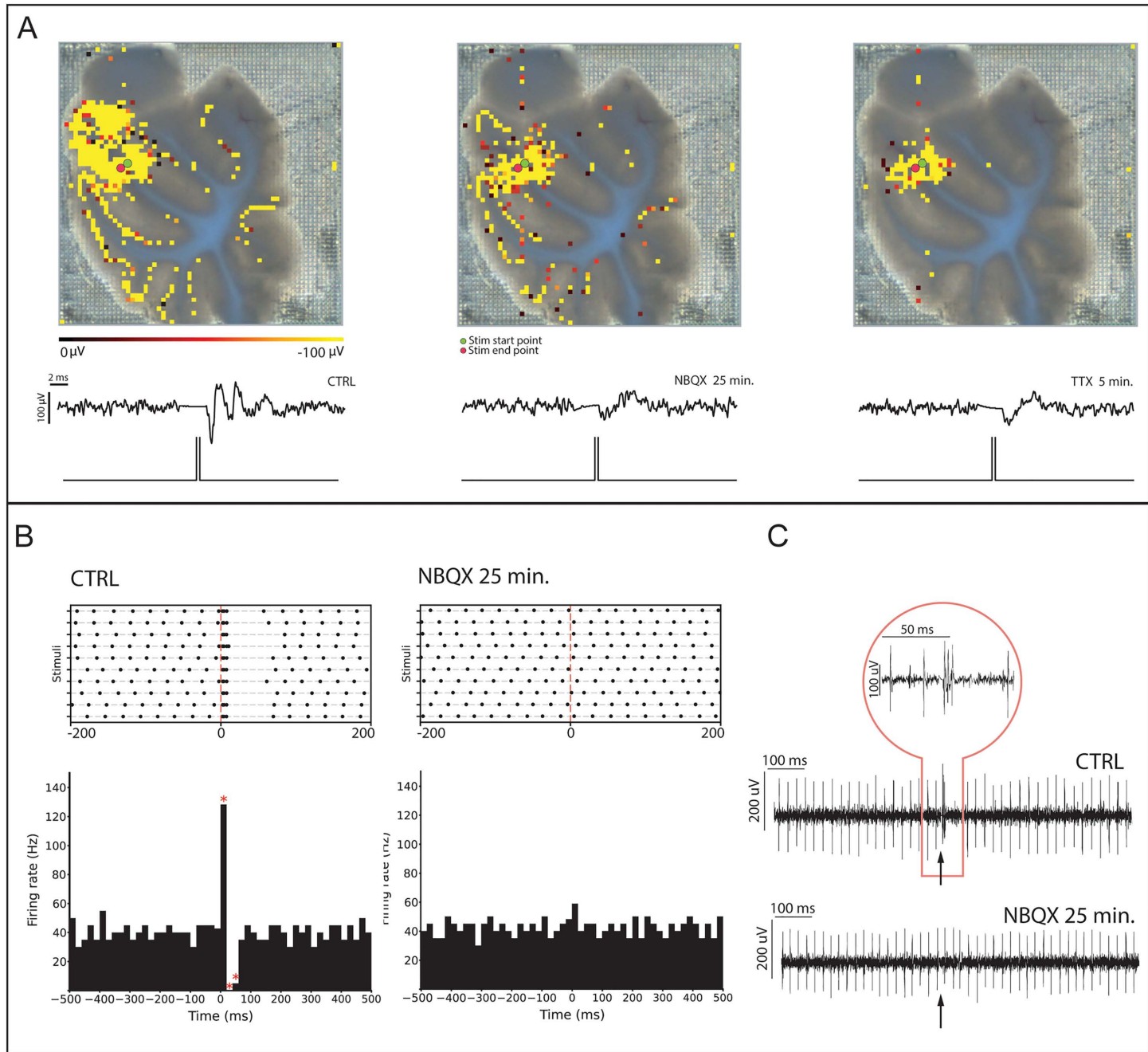

**Fig 6. Neuronal activity evoked by electrical stimulation in cerebellar slices with 3D HD-MEA chips.** A) Stimulation using bidirectional electrodes in correspondence of the mossy fiber bundle evoked granular layer responses in different lobules. The evoked local field potentials (LFP) were abolished by NBQX and TTX. The electrical stimulation is delivered through the channels indicated by red and green dots in the slice image with the activity map superimposed. Color scale bar: 0-300 μV. The single traces refer to LFPs in control condition, 25 min after NBQX perfusion, and 5 min after TTX perfusion. B) Raster plots and peri-stimulus time histograms (PSTH) of a responsive Purkinje cell located in the stimulated lobule in the control condition (CTRL) and after NBQX perfusion. The corresponding raw traces are reported in C). The Purkinje cell burst-pause response follows the synaptic activation of granule cells, indicating that the stimulus effectively propagates through the network. The pause indicates the involvement of feedforward inhibition in the molecular layer.

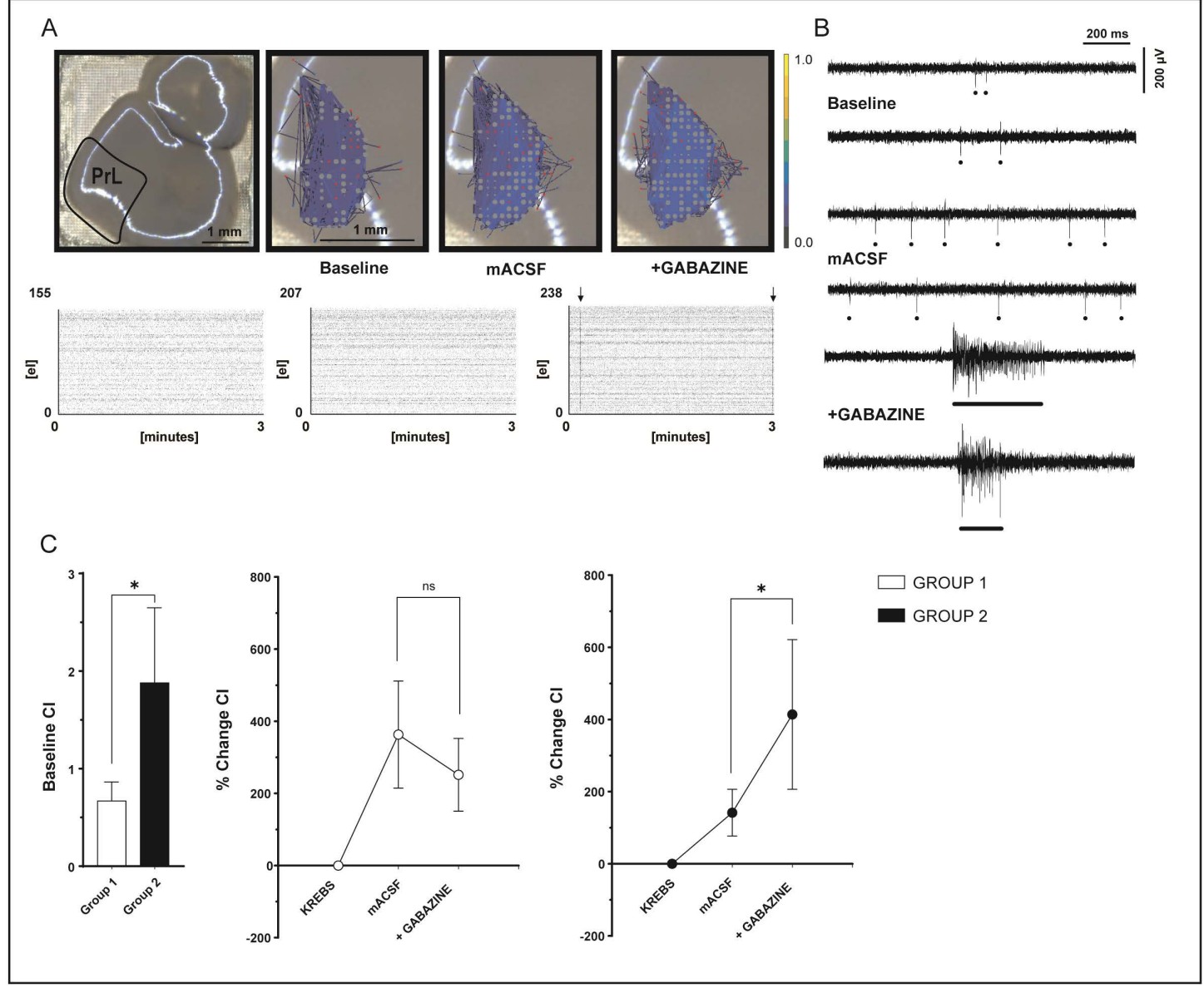

**Fig 7. Characterization of electrical activity from mouse PrL with the 3D HD-MEA.** A) Examples of a brain slice showing LFPs placed on the 3D HD-MEA chip. From left to right: entire slice indicating the PrL location; zoom of the PrL area showing the connectivity maps in Krebs, mACSF, and during gabazine perfusion. The 3 min raster plots of spontaneous activity for each condition are shown below. B) Examples of raw traces of spontaneous activity recordings for each condition (Krebs, mACSF, and gabazine) from the same slice in A. The dots are spike markers. The spontaneous LFP with spikes is indicated with a bar. C) The plots show the correlation index (CI) in Krebs in the two groups of slices (*left*), and the CI % change in the three conditions tested in group 1 (*middle*) and group 2 (*right*). Data are reported as mean ± MSE. Asterisks indicate significant differences at * p < 0.05, ** p < 0.01, *** p < 0.001.

and the regularity of firing (Group 1: CV 0.72 ± 0.02 and CV2 0.89 ± 0.01; Group 2: CV 0.74 ± 0.03 and CV2 0.89 ± 0.01; unpaired Student's t test p = 0.97 and p = 0.99, respectively) were not different (S5 Fig). In both groups, the perfusion of mACSF determined a consistent, though not significant, increase in the basal firing rate. No differences were yet detectable among the two groups in these parameters (group 1: 363 units, basal frequency 0.76 ± 0.05 Hz, CV = 0.74 ± 0.01,

CV2 = 0.90 ± 0.01; group 2: 735 units, basal frequency 0.82 ± 0.10 Hz, CV = 0.75 ± 0.02, CV2 = 0.90 ± 0.01; unpaired Student's t test p = 0.70, p = 0.97, p = 0.99 respectively). The further increase in basal firing rate during gabazine perfusion was significant only for the second group (S5 Fig). Nevertheless, this result might simply reflect the emergence of paroxysmal events, explaining the concurrent change in the CV of the firing of these units (Fig 7A,B). Given the emergence of network events, we calculated the correlation index (normalized on the number of units, CI, see Methods) (Fig 7C) to investigate the connectivity level of the units in the two groups of slices. Interestingly, the correlation analysis revealed that slices in group 1 were characterized by a lower level of network correlation at physiological conditions compared to group 2 (CI group 1 = 0.67 ± 0.19, group 2 = 1.88 ± 0.76, p = 0.08) (Fig 7C). The network correlation significantly increased in mASCF for both groups (group 1: 363.18 ± 148.62%, p = 0.01; group 2: 141.70 ± 64.99, p = 0.02), but further increased after blocking inhibition only in those slices showing paroxysmal events (414.05 ± 207.64%, p = 0.02; Fig 7C). These results show that investigating network properties and emerging phenomena is possible in prefrontal cortex slices using the 3D HD-MEA. Moreover, the electrode high density is also suitable to maintain a good spatial resolution on subdivisions of the PFC, such as the PrL, which often need to be studied separately.

## Measuring activity from brain spheroids with 3D HD-MEA

The need to generate *in vitro* cell cultures capable of maintaining a 3D structure is a challenge that, in recent years, has led to the creation of increasingly complex models such as spheroids and organoids [52,70]. However, such models are typically smaller in size compared to *ex vivo* tissues and have lower cellular density [71,72]. For these reasons, we adopted 3D HD-MEA with thinner electrodes to reduce the potential damage due to needle penetration (Fig 1B). These chips were tested on brain spheroids from primary embryonic neurons, cultured in two different sizes of about 400 μm and 600 μm diameter. The recording area (3.8 by 3.8 mm$^2$) allowed the positioning of multiple samples on the same chip, while keeping a good spatial resolution thanks to the high density of electrodes. Three spheroids of different sizes (large and small) were mounted on two different chips (Fig 8A left panels) showing activity immediately after placement. Acute measurements revealed the presence of robust spiking and bursting activity (Fig 8A right panels). Although the three adjacent spheroids shared the same media, their activity was not synchronized, as evident in the raster plots in Fig 8A (middle panels). Therefore, we considered each sample independent from the others. Large spheroids were penetrated by more μneedles and, consequently, showed a larger number of electrodes detecting active cells (Fig 8B). However, the ratio "active cells/covered electrodes" was similar for both large and small spheroids (~1.5), thus indicating that the sample size did not directly affect the sensing/penetrating capability of the μneedles. The data extracted from each spheroid recording were averaged to generate experimental subgroups sorted by spheroid size. The mean firing rate, bursting rate, and peak-to-peak amplitude of the two groups are shown in Fig 8C. It is interesting to notice that larger spheroids show a trend towards an increased mean firing rate and bursting activity, though not statistically significant, unlike peak-to-peak amplitude (Fig 8C). This suggests that the spheroid size did not impact either the signal amplitude or the neuron-electrode coupling.

## Chemical modulation of organoids activity with 3D HD-MEA

Although brain spheroids are characterized by a 3D architecture, they lack a self-organizing development in a stereotypical cytoarchitecture. This determines a significant lack of complexity in cell composition and neural circuitry [73]. To overcome these limitations, brain organoids are emerging as a novel biological model to investigate human brain development and disorders [74–76]. A 219 days old and 131 days old human iPSC-derived cortical and spinal cord organoids, respectively, were placed on a 3D HD-MEA chip and kept in their medium (see Methods). Acute measurements revealed the presence of hundreds of active sites showing spontaneous spiking and bursting activity in the cortical organoid (Fig 9). The high degree of synchronicity detected (evident in the raster plot of Fig 9B) suggested the presence of network activations. The firing and burst frequency could be modulated by adding 1 mM KCl to the medium (18% and 14% respectively;

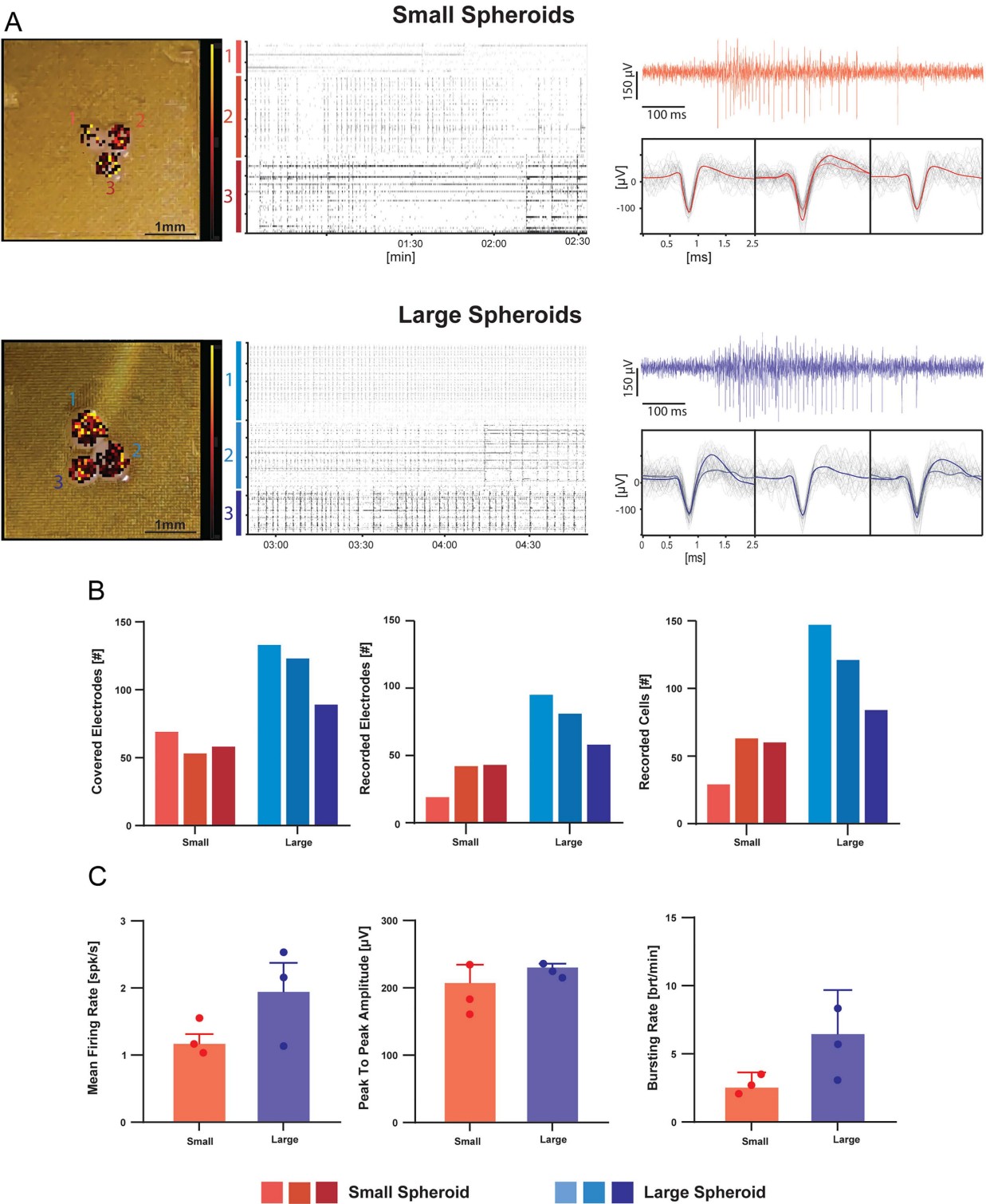

**Fig 8. Recording electrical activity from mouse spheroids with 3D HD-MEA.** A version of 3D HD-MEA equipped with thinner µneedles (width 14 µm, height 65 µm; see Methods and main text) was used with spheroids. A) Pictures of small and large spheroids on chips are overlayed with the corresponding electrical activity maps (*left*) showing the mean firing rate detected by the 3D HD-MEA electrodes (color scale bar: 0-10 spikes/s). The raster plots (*middle*) show the spiking activity over 2.5 minutes. Representative raw traces and averaged waveforms for three electrodes are shown *on*

*the right*. B) The histograms show the number of electrodes sampling the spheroids, the number of active electrodes, and the number of active units for small and large spheroids. C) The histograms show the mean firing rate, the spike peak-to-peak amplitude, and the bursting rate in small and large spheroids. Notice that the spheroid size did not affect the peak-to-peak amplitude. Error bars indicate the standard error.

---

n = 236, p = 0.03) and partially impaired by adding 5 mM (22% and −50% respectively; n = 236, p = 0.007 and p = 0.001 respectively) (Fig 9A,C). Interestingly, the addition of a relatively low KCl concentration was able to alter the synchronicity of the network without significantly modifying the overall spiking activity, while a much higher KCl concentration disrupted network synchronicity with the emergence of tonic firing patterns that tended to disappear in a few minutes [58,59]. Furthermore, KCl perfusion affected the organoid functional connectivity. In particular, the 1 mM KCl increment resulted in a slight, not significant, increased correlation strength and number of links between units (Fig 9D), still preserving the overall topological connectivity map. Conversely, the 5 mM KCl increment led to a strong decrease in the correlation strength and a remodeling of the topological link distribution. Indeed, the correlation value was significantly reduced compared to the previous conditions, and a strong decrease in the number of links was observed (p < 0.0001, −47% and −53%, respectively) (Fig 9D). A different activity pattern was observed on the spinal cord organoid. Basal activity was much less evident with respect to the previous sample. Small KCl concentration increments (+1 mM) resulted in the appearance of synchronous network events that tended to decrease after 4 minutes of compound exposure and completely disappeared with a further increment of KCl concentration (+5 mM) (Fig 10A,B,D). Similar to the cortical organoid, the firing and bursting activity could be modulated by incrementing KCl concentration by 1 mM. The firing and bursting rates increased drastically from 0.06 to 1.2 Hz and from 0.02 to 1.9 bursts per minute, respectively (1916% and 9400%; n = 230, p = 0.0001). Both activity parameters were partially impaired adding 5 mM KCl (66% and 89%, respectively; n = 230, p = 0.0001) (Fig 10C). Interestingly, the mean number of spikes recruited into the burst is significantly larger after the first KCl administration (98%, Fig 10D). Adding 5 mM KCl seemed to regain a spiking level similar to the baseline (Fig 10D). Although the number of spikes per burst remained unchanged between baseline and high KCl concentration, the interval between consecutive spikes within the burst tended to become irregular in the latest phase of the experiment, as indicated by the burst ISI (inter-spike interval) IQR (inter-quartile-range), which provides a measure of ISI regularity within a burst (Fig 10D). The larger heterogeneity of this parameter at high KCl concentration (48%) suggests a disruption of network activity.

## Discussion

In this paper, we demonstrate that 3D HD-MEA is an effective technique for sensing and stimulating interconnected 3D neuronal networks at high resolution while improving tissue vitality and facilitating the pharmacological or electrical modulation of neuronal activity.

### Tissue vitality on 3D HD-MEA

SEM and confocal microscopy assessments, combined with vitality and functional measurements, show that the 3D µneedles penetrate acute slices and spheroids gaining access to the inner neural structures beyond the external cell layer. Moreover, the faster action of active compounds suggests effective diffusion at the base of the chip. This, in turn, correlates well with tissue viability tested with calcein staining. A possible concern of penetrating 3D chips is the perturbation of tissue integrity caused by the µneedles, given the thousands of µneedles in a few mm² (4096 electrodes in a 3.8 by 3.8 mm² area). VSDi on cerebellar slices did not reveal alterations in cerebellar network activation on 3D HD-MEA chips compared to data obtained from a slice positioned on a coverslip. Furthermore, measurements from both spheroids and organoids showed strong sustained network burst activity compatible with good tissue vitality. Taken together, these results indicate that 3D HD-MEA chips, compared to planar MEAs, increase tissue vitality and provide efficient tissue penetration without visibly affecting neuronal network structure and function.

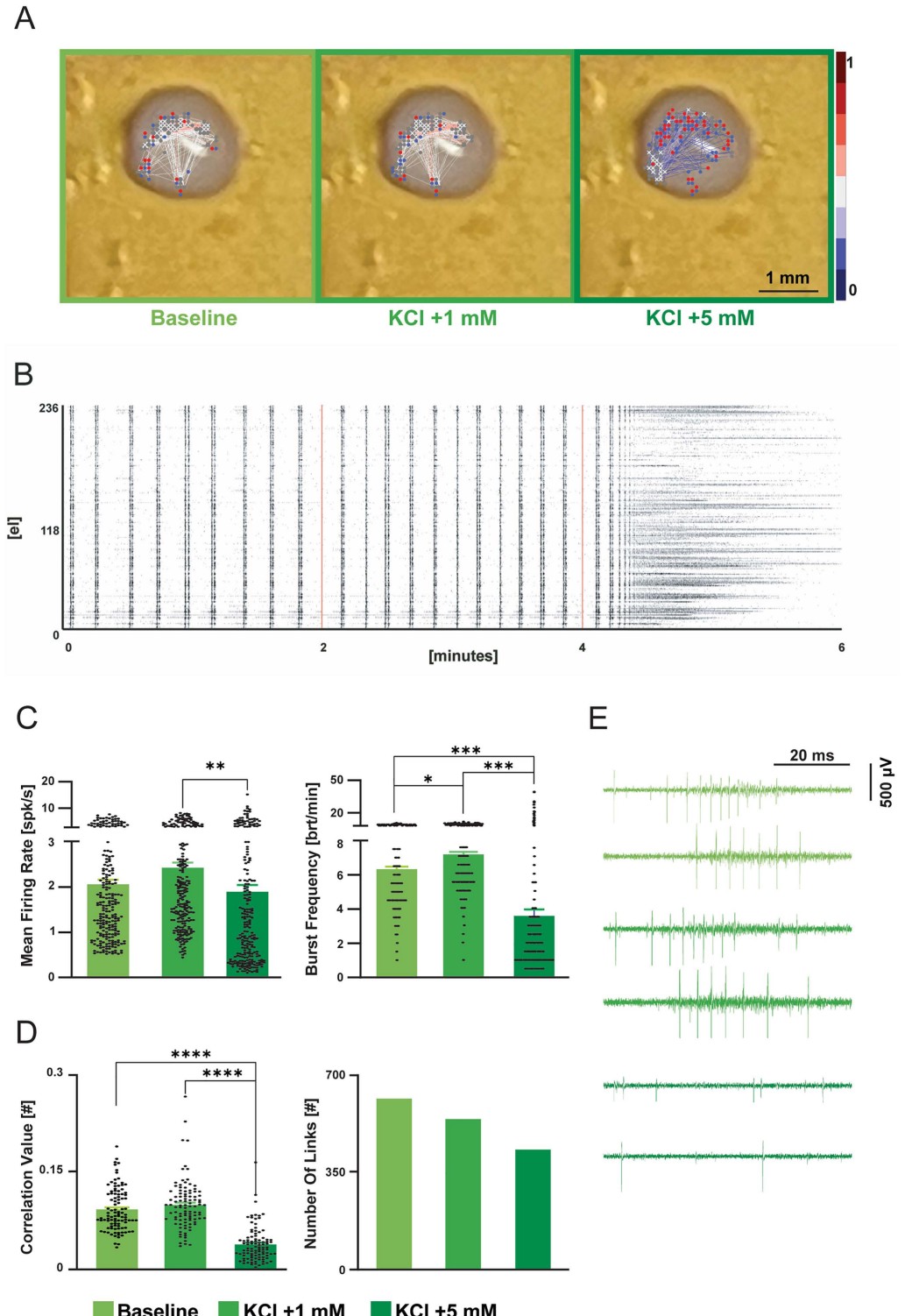

**Fig 9. Recording electrical activity from a human cortical organoid with 3D HD-MEA.** A) Pictures of an organoid on chip are overlayed with the corresponding connectivity map (from left to right: baseline, + 1 mM, + 5 mM KCl). Connectivity maps are calculated using cross-correlation analysis (see Methods). Each link is color-coded based on its normalized correlation level. Red and blue dots represent the nodes with incoming and outgoing links respectively, grey dots represent nodes with both links. B) Raster plot indicating the activity pattern over time for each experimental condition. Black dots

indicate the detected spikes, red lines indicate the change in [K$^+$] in the medium. C) The histograms show the firing and bursting metrics to determine activity and network synchronicity, respectively. D) The histograms show the strength and number of links, resulting from cross-correlation analysis. Error bars indicate the standard error. E) Representative traces for each recording phase showing the modulation of spiking activity. Asterisks indicate significant differences at * $p < 0.05$, ** $p < 0.01$, *** $p < 0.001$.

### Recording efficiency with 3D HD-MEA

Besides improving tissue vitality, the 3D HD-MEA improved signal acquisition efficiency, as revealed by the increased number of active electrodes, the cells to electrodes ratio, and the number of recorded units. Moreover, the electrical stimulation of 3D tissue by penetrating µneedles was more efficient than with planar MEAs. Consequently, the 3D HD-MEA chips could be used to stimulate the neuronal microcircuit and record its response in different areas opening the possibility of comparing network and subnetwork properties in the same brain slice. When tested on challenging samples, the 3D chip detected activity that proved previously difficult to trace. In particular, the system was sensitive enough to detect the sparse basal firing activity of neurons in the prelimbic region of the medial PFC without the need for increasing their excitability. Furthermore, when large network events occurred, both the low-frequency (LFP) and high-frequency (spikes) components of electrophysiological signals could be detected.

### Recordings from acute PFC slices

Here, we show that the PFC activity can be detected in physiological conditions using the 3D HD-MEA and that complex network properties can be analyzed in these recordings, including pharmacological modulations of activity. The ability to record from large proportions of neurons is critical to investigating cortical processing, which is mostly based on distributed network activity and synchronization. While the organization and coding of neuronal activity have been investigated in brain regions like the motor cortex, a characterization of PFC neuron activity in physiological conditions remained elusive mostly due to the difficulty of recording spontaneous activity in acute PFC slices using electrophysiological techniques [77–79]. Indeed, planar MEA recordings provided almost only local field potentials [80] and required manipulation of the extracellular medium to increase neuronal excitability [81,82], while calcium imaging could not record single cell spikes directly [83,84].

The 3D HD-MEA allowed to resolve the low firing rate of the PFC [85–90], also due to the need for a glutamatergic drive [91,92], at the same time preventing from using non-physiological conditions such as 0mM Mg$^{2+}$ solutions used to increase neuronal excitability [93,94]. Interestingly, our data allowed the quantification of the correlated activity of multiple units in physiological conditions, providing a measure of the connectivity and strength of the PrL network in slice. Importantly, the high density of electrodes allowed sampling of the PrL activity selectively, confining the analysis to a subregion of the PFC and maintaining many channels available for the statistical validity of the results (though less than half of the channels sampling the PrL were active, the number of units recorded in 7 slices was 983).

### Recordings of spheroids and organoids

3D HD-MEAs proved suitable also for recording neural spheroids and organoids. It is noteworthy that brain spheroids have a lower cellular density compared to brain slices and a smaller size (usually a few hundred µm), making it challenging to record from multiple sites in the same sample. Therefore, the smaller µneedle size used for these recordings was probably crucial. Interestingly, multiple spheroids could be recorded on the same 3D HD-MEA, opening the possibility of testing several specimens simultaneously. Concerning organoids, the ability to perform acute measurements without the need to culture them on-chip for several days avoids the effects of cell migration typically seen on planar MEAs. This offers two main advantages: first, it ensures that neurons are recorded in a 3D environment rather than

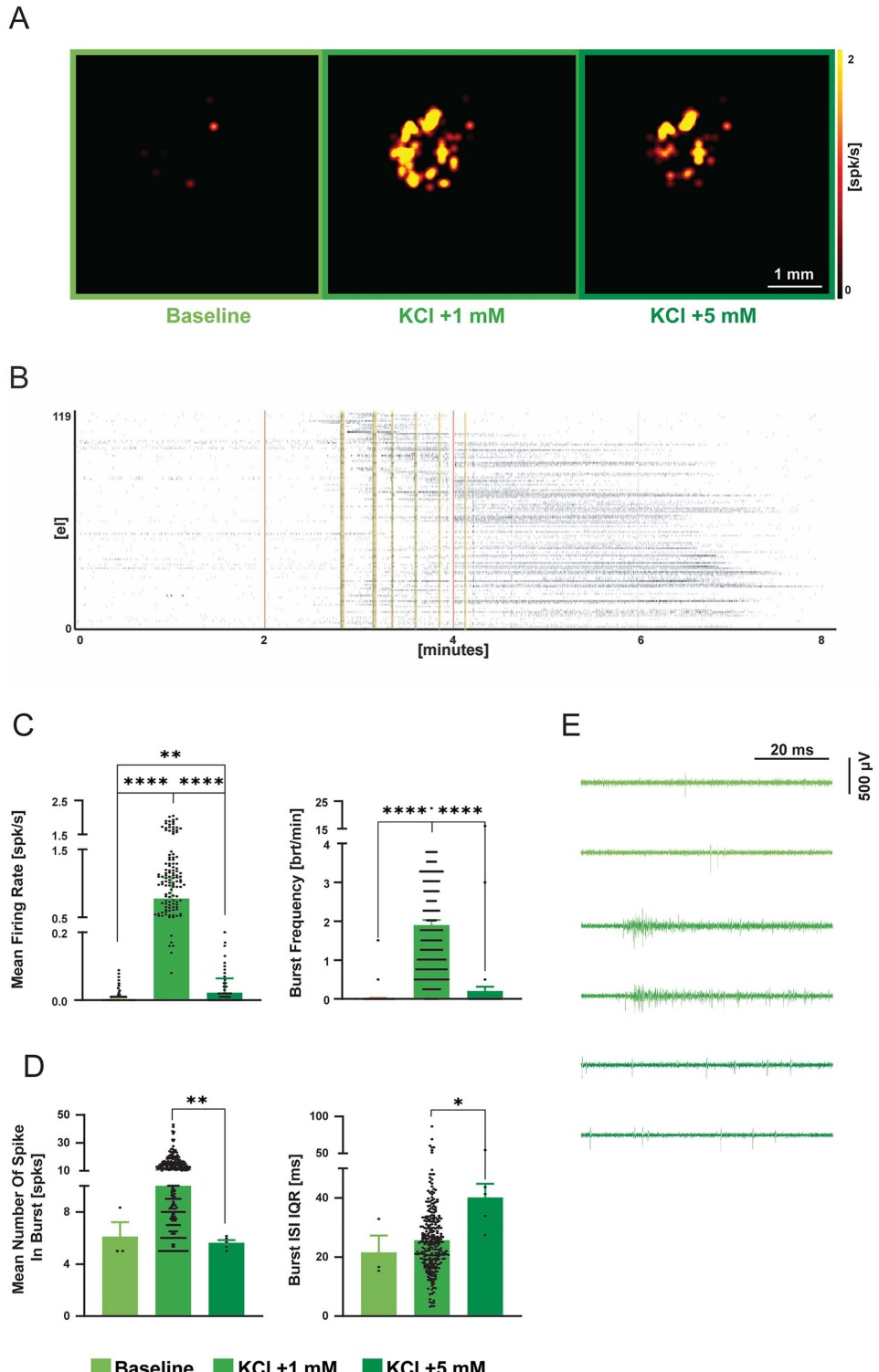

**Fig 10. Recording of electrical activity from a human spinal organoid with 3D HD-MEA.** A) Spike maps indicating the color-coded mean firing rate for each electrode of the chip in the three conditions tested (from left to right: baseline, + 1 mM, + 5 mM KCl). B) Raster plot indicating the activity pattern over time of each experimental condition. Black dots indicate the detected spikes. Red lines indicate the change in [K+] in the medium; bursts are highlighted in yellow. C) The histograms show the firing and bursting metrics to determine activity and network synchronicity. D) Raising KCl concentration of

1 mM resulted in an increased, although not significant, number of spikes recruited in bursts. Conversely, a higher KCl concentration (+5 mM) caused the number of burst spikes to recover baseline (p < 0.0001, n = 230) and to increase the ISI irregularity within bursts (p < 0.05, n = 230). Error bars indicate the standard error. E) Representative traces for each recording phase showing the modulation of spiking activity. Asterisks indicate significant differences at * p < 0.05, ** p < 0.01, **** p < 0.0001.

the 2D network formed due to cell migration; second, it prevents alterations of the 3D structures or the increase of the necrotic core, which can occur when organoids, after growing in a free-floating environment, are immobilized. Moreover, we could modulate the firing frequency and synchrony in cortical and spinal organoids, in which such manipulations are usually difficult to achieve. Indeed, KCl-induced membrane depolarization is known to efficiently modulate neural connectivity in both vertebrate and invertebrate primary cultures [95,96], but such modulation is yet to be characterized in brain organoids.

## Conclusions

The 3D HD-MEA technology raises the efficiency of electrophysiological recordings and stimulation in 3D biological models. We think this work is preliminary to answer specific experimental questions that were left open before. For example, a high resolution of recording and stimulation sites is needed to map the distribution of electrical signals in brain slices, e.g., of the cerebral cortex, cerebellum, and hippocampus, and to address unresolved physiological and computational questions. The 3D HD-MEA also offers the opportunity to explore neural activity in spheroids and organoids grown and maintained in physiological conditions, contributing to reducing the gap between observations in animal and human tissue. In all these preparations, the action of drugs applied through the perfusion bath would benefit from the faster and more complete exchange of fluids and molecules. Thus, combined, the advantages offered by the 3D HD-MEA would allow us to investigate the spatiotemporal dynamics of neuronal activity and synaptic plasticity and the neuromodulator effect of drugs. Finally, the massive amount of data obtained from simultaneous recordings of thousands of interconnected neurons from the same sample is ideal for instructing computational models of neuronal networks, gaining mechanistic insight into circuit organization and neural dynamics [97].

## Supporting information

**S1 Fig. Stability of the number of active channels in planar and 3D HD-MEA chips.**
(TIF)

**S2 Fig. Characterization of the amplification performance of planar vs.3D HD-MEA chips.**
(TIF)

**S3 Fig. Neuronal activity evoked by electrical stimulation in cerebellar slices with 3D HD-MEA chips.**
(TIF)

**S4 Fig. Characterization of PrL activity in slices without LFPs using 3D HD-MEA chips.**
(TIF)

**S5 Fig. Characterization of spontaneous firing of PrL activity units with 3D HD-MEA chips.**
(TIF)

**S1 Movie. Images stack at different focal planes show μneedle penetration into a brain spheroid mounted on a 3D HD MEA.** Scale bar 180μm, magnification 50X.
(AVI)

**S2 Movie. Comparison of the TTX effect on cerebellar slices mounted on a planar and 3D HD-MEA.** Faster activity decrease on the 3D HD-MEA can be observed both on the activity map and representative raw traces.
(MP4)

**S3 Movie. Electrical stimulation of the granular layer in a cerebellar slice.** Activity map and representative raw traces show the evoked response.
(MP4)

**S4 Movie. Spontaneous local field potential (LFP) originating at the PrL area and spreading through the entire slice, recorded in mACSF with gabazine.** Representative raw traces show the LFP and spiking activity characterizing the paroxysmal activity.
(MP4)

## Acknowledgments

The authors want to thank Dr. Michele Dipalo at the Italian Institute of Technology (IIT, Genova Italy) in providing SEM images of a cerebellum slice on a 3D HD-MEA chip shown in Fig 1D. The authors also thank members of the Emory Brain Organoid Hub A. King, Z. Ou and R. Yu for providing organoids. The authors gratefully acknowledge Centro Grandi Strumenti and the Confocal Microscopy Facility for their support and assistance in this work; the authors also thank Patrizia Vaghi of Centro Grandi Strumenti Confocal Microscopy Facility for her support and assistance in this work.

## Author contributions

**Conceptualization:** Lisa Mapelli, Mariateresa Tedesco, Simona Tritto, Mauro Gandolfo, Steven A. Sloan, Alessandro Maccione, Egidio D'Angelo.

**Formal analysis:** Lisa Mapelli, Danila Di Domenico, Giacomo Sciacca, Francesco Mainardi, Simona Tritto, Alessandro Maccione.

**Funding acquisition:** Alessandro Maccione, Egidio D'Angelo.

**Investigation:** Lisa Mapelli, Danila Di Domenico, Giacomo Sciacca, Francesco Mainardi, Alessandra Ottaviani, Anita Monteverdi, Mariateresa Tedesco, Chiara Rosa Battaglia, Simona Tritto, Kilian Imfeld, Stefanie Kiderlen, Lukas Krainer, Chiara Cervetto, Manuela Marcoli, Anson Sing, Jimena Andersen, Fikri Birey, Alessandro Maccione.

**Methodology:** Lisa Mapelli.

**Project administration:** Lisa Mapelli.

**Supervision:** Lisa Mapelli, Alessandro Maccione.

**Validation:** Lisa Mapelli.

**Visualization:** Lisa Mapelli, Danila Di Domenico, Giacomo Sciacca, Francesco Mainardi.

**Writing – original draft:** Lisa Mapelli, Danila Di Domenico, Giacomo Sciacca, Mariateresa Tedesco, Kilian Imfeld, Alessandro Maccione.

**Writing – review & editing:** Lisa Mapelli, Alessandro Maccione, Egidio D'Angelo.

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
