## [Decision Letter · Decision Letter 0]

24 Mar 2025

PONE-D-25-08619Enhanced electrophysiological recordings in acute brain slices, spheroids, and organoids using 3D high-density multielectrode arraysPLOS ONE

Dear Dr. Mapelli,

Thank you for submitting your manuscript to PLOS ONE. After careful consideration, we feel that it has merit but does not fully meet PLOS ONE’s publication criteria as it currently stands. Therefore, we invite you to submit a revised version of the manuscript that addresses the points raised during the review process.

Both reviewers feel that additional validation is needed and wrote that out in the feedback. Reviewer 1 would like to see more validation between 2D vs 3D chips while reviewer 2 needs some clarification about the experimental conditions. Overall, the suggestions are fair and can be adressed in due time. 

We look forward to receiving your revised manuscript.

Kind regards,

Gerrit Hilgen

Academic Editor

PLOS ONE

Journal Requirements:

2**.** To comply with PLOS ONE submissions requirements, in your Methods section, please provide additional information regarding the experiments involving animals and ensure you have included details on (1) methods of sacrifice, and (2) efforts to alleviate suffering.

“The authors acknowledge the #NEXTGENERATIONEU (NGEU) and Ministry of University and Research (MUR), National Recovery and Resilience Plan (NRRP), project MNESYS (PE0000006) – A Multiscale integrated approach to the study of the nervous system in health and disease (DN. 1553 11.10.2022) to ED. The authors acknowledge the European Union – Next GenerationEU – National Recovery and Resilience Plan (NRRP) – Mission 4 Component 2 Investiment 1.1 Call PRIN 2022 PNRR CUP  F53D23010320001 (MUR code P2022YMM29 - Functional and computational investigation of brain networks in PCDH19-related developmental and epileptic encephalopathy-9. A close-up on Parvalbumin interneurons) to LM.”

“European Union’s Horizon Europe Programme under the Specific Grant Agreement No. 101147319 (EBRAINS 2.0 Project) to ED (https://www.ebrains.eu/).

European Union’s Horizon 2020 research and innovation programme under the grant agreement 964877 – NEUCHIP to AMa (https://neuchip.eu/).

The sponsors did not play any role in the design, data collection, analysis, decision to publish, or preparation of the manuscript.”

“Mariateresa Tedesco, Giacomo Sciacca, and Chiara Rosa Battaglia are employers of 3Brain AG.

Mauro Gandolfo, Kilian Imfeld, and Alessandro Maccione are shareholders of 3Brain AG.

The other authors declare that the research was conducted in the absence of any commercial or financial relationships that could be construed as a potential conflict of interest.”

Reviewers' comments:

Reviewer's Responses to Questions

**Comments to the Author**

1. Is the manuscript technically sound, and do the data support the conclusions?

Reviewer #1: Partly

Reviewer #2: Yes

2. Has the statistical analysis been performed appropriately and rigorously? 

Reviewer #1: Yes

Reviewer #2: Yes

3. Have the authors made all data underlying the findings in their manuscript fully available?

Reviewer #1: Yes

Reviewer #2: Yes

4. Is the manuscript presented in an intelligible fashion and written in standard English?

Reviewer #1: Yes

Reviewer #2: Yes

5. Review Comments to the Author

Reviewer #1: Mapelli et al. provides a methods paper for the use of a 3D high-density multielectrode array. The authors provide interesting and helpful insights on how the 3D high-density array compares to the standard planar arrays. They showcase how the 3D array might improve data collection with the 3Brain BioCam using the 3D chips over the 2D chips; a manuscript I can support being published but the following points should first be fully considered. Some of the below points can aid in the claim that the 3D arrays offer “enhanced electrophysiology”.

Major points

1. Authors should provide some characterization that compares electrophysiological “noise” between the 2D and 3D chips.

2. Figure2A – It would be more appropriate for comparison of tissue viability to directly compare the 2D and 3D chip, not just compare the 3D chip to a slice on a glass coverslip.

3. Figure 4 - Authors should consider spike sorting to determine if there is an increased number of neurons detected from the cortico-hippocampal slices. Also, spike sorting the data from the cortico-hippocampal slices for fast-spiking interneurons would be informative. Is there a greater probability of recording from FS-interneurons with the 3D chips? Data is all collected, and this would be just additional analysis.

4. Data comparing LFP signals from the 2D and 3D chips would be very informative and straightforward with the already collected data. Do the 3D chips provide readout of greater power in the low frequency bands (1-30Hz) compared to the 2D chips? Can these new 3D chips allow for greater comparisons of high-frequency oscillations between samples (ripples and fast-ripples)? These LFP limitation of the 2D chips were outlined recently in the last two figures here: https://app.jove.com/t/67065/high-quality-seizure-like-activity-from-acute-brain-slices-using. Do the 3D chips offer any improvement?

Minor points

1. Line 113 – replace killed with euthanized

2. Line 160 – define hiPSCs

3. Line 72 – change intricated to intricate

4. Line 139 – change neurons to neuron

5. Line 140 – change cortexes with cortices

6. Line 407 – fix grammar “which allows to detect neuronal membrane”

7. Line 422 – add “=” in Scale bar 40um.

Reviewer #2: Dear editor,

Regarding the manuscript titled ‘Enhanced electrophysiological recordings in acute brain slices, spheroids, and organoids using 3D high-density multielectrode arrays’ by Mapelli et al. The paper is excellently written, very thorough in details and provides a report on an important progress in the field of microelectrode arrays, addressing multiple common limitations found in the most commonly used planar MEAs arrays.

This is achieved by using a 3D high density micro electrode array (3D HD-MEA), which has several advantages over planar arrays. The authors highlight that planar arrays, while important in several instances have some key drawbacks, including:

- Planar arrays can only record signals which mostly originate from the outer layers, limiting the access to the dynamics of cells inside any 3D structure.

- Due to hardware limitations, it is challenging to record from multiple sites over large areas.

- Due to the intrinsic nature of how cells attach to a planar array, it imposes several difficulties regarding nutrient delivery, oxygenation and waste diffusion to avoid necrosis in the core of the tissue.

The authors main claims regarding the advantages of 3D HD-MEA are the following:

- The 3D HD-MEA microneedles reached the inner layers of samples without damaging network integrity.

- The microchannel network between microneedles improved tissue vitality and chemical compound diffusion.

- Signal recording and stimulation efficiency proved higher with the 3D HD-MEA than with a planar MEA.

- 3D HD-MEA resolved the challenge of recording from brain spheroids as well as cortical and spinal organoids.

Overall, I am very satisfied with the quality of the manuscript, and I consider it to be of great value to the scientific community, particularly due to the ability of this new technology to address or make use of biological preparations which are closer to the ones found in in-vivo work and to balance the advantages of high-density MEA while minimizing the damage to the tissue. Tests were intuitive, well-reasoned and the plots are appropriate and clear, the supplementary material is adequate and expands upon points that are of great use for specialists in the field and all the references seem logical and relevant.

Having said that, I do have some points that I believe are worthy of being addressed, both major and minor. None of my concerns demand extra experiments, but rather a more detailed explanation of the reasoning behind certain techniques and their consequences in comparing different arrays. I have divided my concerns into major and minor issues. Please see a detailed explanation below:

Major issues

1) Lines 436-439, it reads ‘The slices were first placed on a planar HD-MEA chip and then transferred onto a 3D HD-MEA chip (or vice versa) and recorded for 3 minutes in each case. The 3D HD-MEA chip showed increased performance in detecting activity from the cerebellar Purkinje cell layer compared to the planar HD-MEA…’. One of the main claims of this manuscript is that the 3d HD-MEA has a marked improvement in signal recording capabilities over planar MEA. For this, the authors compare the recording quality of the same cerebellar brain slice in a planar MEA and a 3D HD-MEA. As described in the methods (lines 258-261), there some methodological differences between the two recording arrays, namely that the planar array makes use of a platinum anchor, and that the activity in the brain slices was recorded 5 minutes after the slice was positioned on the chip.

My main concern is that using a planar array, 5 minutes may not be sufficient for the brain slice to achieve a good adhesion to the contacts on the array, thus, decreasing the recording quality of the planar array.

My understanding is that an ideal time is between 10-30 minutes before starting any recording; 5 minutes is going towards the bare minimum and this depends on the quality of the brain slices. I take it that the authors did their due diligence to ensure the highest possible quality of brain slices, and while I don’t consider that further experiments are necessary, I would encourage the authors to comment on their reasoning behind the timing for the planar arrays as well as the tradeoffs that occur when transplanting the same brain slice into the 3D HD-MEA.

In a way, the fact that more units and the overall yield is increased in 3D HD-MEA is in further support of the authors main arguments; but on the other hand, a naïve counter argument could be that the mechanical stress that the slices are subjected to could make the very same brain slice more easily accessible for the ‘needles’ in the 3D HD-MEA, increasing the number of units recorded and the overall yield. Thus, making the comparison between the planar and 3D arrays more difficult to interpret, since its possible that the mechanical stress that the brain slice is subjected to, may actually help the microneedles to penetrate the brain slice.

This concern is not addressed in any of the supplementary materials, and thus I bring it to attention. I would recommend that the authors expand their reasoning on the methods portion of the manuscript, section 2.6 ‘Electrophysiological recordings’.

2) Numbers in lines 442-448, which make reference to Figures 3B and 3C are odd. The figures describe absolute number of electrodes and/or cells. According to the methods, the number of cerebellar slices were 6. But the results described on the lines refer to percentages, but I believe that these descriptive statistics using percentages are not contributing to highlight the actual number of electrodes, number of cells and the ratio of cells/electrode. Overall, the same sentiment applies for Figure 4. I’d appreciate an explanation by the authors as to why they consider using percentages on the main text, while using (what I believe to be) absolute numbers in the figure. I think that it would be clearer if they used absolute numbers in the text and explain the spread of the values due the variability of one slice to the next. Perhaps is me being sleep deprived, but its very much non intuitive to link the results from Figure 4B, a histogram with 15 bins to the claim of lines 462-464 ‘The 3D HD-MEA showed a significantly larger number of active electrodes in the cortical area compared to the planar HD-MEA (+371±46 %, n=3 slices, p = 0.0384)’. Again, I apologize if I completely misunderstood, but the link between the histogram and the +371±46 % is not clear and I think that the data to support the claim is there, but I also think that there is a better, clearer way to express these results.

3) Lines 651-653, refer to ‘links’ found in the preparation involving the prefrontal cortex. While in the methods, the authors make reference to using BrainWave 5 and some custom python scripts, I think this claim requires at least a reference that explains what these links are and how they’re obtained. There are references provided for other features in line 316, but not for this issue, so I think this needs to be explained more clearly and referenced.

Minor issues

These are for the most part my personal preference, and I understand we may have different opinions on the manner. The manuscript is very well explained and approachable, but I think these minor corrections improve the writing in some instances:

• Line 83, “In 2023, an advanced generation of 3D HD-MEAs was introduced to the scientific community”, please add specific reference. The manuscript places references to how HD-MEAs have been validated, and it seems like a continuation of the work by Wang et al. (37), but it’s unclear if it’s the same work or something else. Please consider revising this sentence.

• Line 358, reads ‘…and used scansion electron microscopy’. Correct it to ‘and used scanning electron microscopy’

• Line 384-385 reads ‘we performed vitality and functionality tests on three different microchannel configurations’. Tissue vitality is one of the main claims of this work, and I consider the tests made for testing it where adequate; however, this paper has the potential to be read by people who are working with live animal models and thus, the link between the dimensions of microchannel configuration/sizes and tissue vitality may not be intuitive or straightforward. I encourage the authors to briefly state what links these two parameters, in order to make more accessible the contents of the manuscript. This is succinctly explained on lines 403-404, but I think it improves readability by addressing why channel size is important in diffusion right before testing different sizes of channel configuration.

• Line 412, missing a closing parenthesis.

• The colors for figure 9C, D and E could be better, especially for 9E, since the traces with white background are difficult to see. I think darker shades of green would be better. Same for figure 10C, D and E.

Again, I congratulate the authors for this work, and I encourage the publication of this manuscript as long as the major issues are addressed (or explained in the case of the percentages) and corrected. I believe my observations and recommendations are practical, and overall, it would improve the readability of the manuscript.

Thank you for your consideration, wishing you the very best,

6. PLOS authors have the option to publish the peer review history of their article (what does this mean? ). If published, this will include your full peer review and any attached files.

**Do you want your identity to be public for this peer review?** For information about this choice, including consent withdrawal, please see our Privacy Policy .

Reviewer #1: No

Reviewer #2: **Yes: ** Luis Fernando Cobar Zelaya

---

## [Author Response · Author response to Decision Letter 1]

11 Jun 2025

Reviewer 1

Mapelli et al. provides a methods paper for the use of a 3D high-density multielectrode array. The authors provide interesting and helpful insights on how the 3D high-density array compares to the standard planar arrays. They showcase how the 3D array might improve data collection with the 3Brain BioCam using the 3D chips over the 2D chips; a manuscript I can support being published but the following points should first be fully considered. Some of the below points can aid in the claim that the 3D arrays offer “enhanced electrophysiology”.

We thank the Reviewer for the positive comments, which helped improve the manuscript. We addressed all the issues raised. A point-by-point reply is provided below.

Major points

1. Authors should provide some characterization that compares electrophysiological “noise” between the 2D and 3D chips.

We thank the reviewer for the suggestion. While calculating the electrophysiological “noise” level is surely interesting, we believe that, in this case, it is even more valuable to assess the amplification capabilities of the planar vs. 3D HD-MEAs circuitries, to evaluate the chip performance. A direct comparison of the electronics' signal-to-noise ratio (SNR) would provide a more robust interpretation of our data. To achieve this, we calculated and compared the SNR across a group of five planar and five 3D HD-MEAs using a synthetic sinusoidal signal in dry conditions, observing no significant differences. These results confirm that the amplification performance is consistent regardless of HD-MEA chip type (as expected since the chip circuitries for both planar and 3D HD-MEAs are identical), thus demonstrating that the advantages in terms of electrode sensitivity are given by the µneedles fabricated on top of the electrodes. This characterization is now reported in Supplementary Figure 2 and described in Methods in the new section “Characterization of Electronic Chip Performance”.

2. Figure2A – It would be more appropriate for comparison of tissue viability to directly compare the 2D and 3D chip, not just compare the 3D chip to a slice on a glass coverslip.

We repeated the calcein viability experiments on slices placed on planar chips, following the suggestion. This also allowed us to increase the control sample compared to slices placed on 3D HD-MEA chips. Since the results on planar chips are more significant in this context, we have directly reported this data in the revised text and figure. Although slightly increased, the vitality result on 2D chips is not significantly different compared to the previous control on coverslips (38% compared to 33%). The revised control is still considerably different compared to the vitality shown by slices placed on 3D chips with large microchannels, confirming the increased vitality considerations in the original manuscript. These new data have been added to the revised manuscript and reported in Figure 2.

3. Figure 4 - Authors should consider spike sorting to determine if there is an increased number of neurons detected from the cortico-hippocampal slices. Also, spike sorting the data from the cortico-hippocampal slices for fast-spiking interneurons would be informative. Is there a greater probability of recording from FS-interneurons with the 3D chips? Data is all collected, and this would be just additional analysis.

We thank the Reviewer for these useful comments.

We performed spike sorting to assess potential differences in the number of recorded cells between planar and 3D HD-MEAs, as suggested. As expected, spike sorting changed the numerical values of previously extracted metrics, such as the “number of active electrodes”, “number of spikes”, and “peak-to-peak amplitude”. However, the overall trend of increased activity detected by the 3D HD-MEA remained unchanged, and a general trend toward a higher number of detected cells with the 3D HD-MEA compared to the planar configuration is still evident, though not statistically different. It is important to note, however, that extracellular recordings from cortical slices typically yield cells with a relatively low spiking frequency, which may limit the reliability of spike sorting. We believe that the lack of statistical significance in our findings may be attributed to the limited reliability of spike sorting in identifying single units in this condition.

In studies aimed at identifying fast-spiking interneurons, most existing research (e.g., Wang et al., 2016; Markram et al., 2004) primarily relies on patch-clamp techniques, where these interneurons are distinguished based on their intracellular responses to current injections at the single-cell level. Only a limited number of studies have employed MEA systems to differentiate neuronal subpopulations, such as cortical fast-spiking interneurons, by analyzing their extracellular activity. For instance, Becchetti et al. (2012) investigated primary cortical neuron cultures using various metrics commonly applied to characterize fast-spiking interneurons, including the coefficient of variation, inter-spike interval, autocorrelation, and Fano factor. Among these, the Fano factor provided the clearest distinction between regular-spiking (putative pyramidal) neurons and fast-spiking (putative inhibitory) neurons. Following this approach, we calculated the Fano factor (using a 6-second time window) for all active units recorded in the cortex. Previous studies (Tan et al., 2014; Nawrot et al., 2008) suggest that fast-spiking interneurons in ex vivo brain slices typically exhibit a Fano factor between 1 and 1.5. We then quantified the number of units recorded using both planar and 3D HD-MEA that fell within this range and observed no significant difference between experimental conditions.

Figure 1: Putative fast spiking neurons detected by counting all the units displaying a Fano Factor within 1 and 1.5. No significant difference has been found. Error bars indicate the standard error.

However, a more reliable identification of neuronal phenotypes would require morphological analysis or immunohistochemistry. Therefore, given the limitations of this approach, we did not include this further characterization in the manuscript.

• Markram H, Toledo-Rodriguez M, Wang Y, Gupta A, Silberberg G, Wu C. Interneurons of the neocortical inhibitory system. Nat Rev Neurosci. 2004 Oct;5(10):793-807. doi: 10.1038/nrn1519. PMID: 15378039.

• Wang B, Ke W, Guang J, Chen G, Yin L, Deng S, He Q, Liu Y, He T, Zheng R, Jiang Y, Zhang X, Li T, Luan G, Lu HD, Zhang M, Zhang X, Shu Y. Firing Frequency Maxima of Fast-Spiking Neurons in Human, Monkey, and Mouse Neocortex. Front Cell Neurosci. 2016 Oct 18;10:239. doi: 10.3389/fncel.2016.00239. PMID: 27803650; PMCID: PMC5067378.

• Becchetti A, Gullo F, Bruno G, Dossi E, Lecchi M, Wanke E. Exact distinction of excitatory and inhibitory neurons in neural networks: a study with GFP-GAD67 neurons optically and electrophysiologically recognized on multielectrode arrays. Front Neural Circuits. 2012 Sep 6;6:63. doi: 10.3389/fncir.2012.00063. PMID: 22973197; PMCID: PMC3434456.

• Tan AYY, Chen Y, Scholl B, Seidemann E, Priebe NJ. Sensory stimulation shifts visual cortex from synchronous to asynchronous states. Neuron. 2014 Apr 16;82(4):936-948. doi: 10.1016/j.neuron.2014.03.019. PMID: 24742464; PMCID: PMC4006717.

• Nawrot MP, Boucsein C, Rodriguez-Molina V, Aertsen A, Grün S, Rotter S. Measurement of variability dynamics in cortical spike trains. J Neurosci Methods. 2008 May 30;169(2):374-390. doi: 10.1016/j.jneumeth.2007.10.013. PMID: 18093641.

4. Data comparing LFP signals from the 2D and 3D chips would be very informative and straightforward with the already collected data. Do the 3D chips provide readout of greater power in the low frequency bands (1-30Hz) compared to the 2D chips? Can these new 3D chips allow for greater comparisons of high-frequency oscillations between samples (ripples and fast-ripples)? These LFP limitation of the 2D chips were outlined recently in the last two figures here: https://app.jove.com/t/67065/high-quality-seizure-like-activity-from-acute-brain-slices-using. Do the 3D chips offer any improvement?

We appreciate the Reviewer’s insightful suggestion.

To approach the issue, we performed a spectral analysis with careful selection of events for FFT processing. Specifically, we analyzed hippocampal events extracted from the same region of the same slice (n=2), first recorded with a 2D chip and then with a 3D chip. This preliminary analysis revealed notable differences in the 60–150 Hz frequency band (see figure below). While these results are encouraging, we acknowledge that they should be interpreted with caution. Moreover, our experiments were conducted under non-physiological conditions (high K⁺ and Mg²⁺-free), which induce dynamic changes in signal frequency and intensity rather than a stable, linear field potential activity. Therefore, these conditions are not optimal for the type of comparison suggested. A thorough investigation would require a dedicated experimental design and a larger sample size to minimize variability, efforts that fall beyond the scope of the present study. Indeed, a rigorous comparison of the sensitivity of 2D and 3D chips in detecting low-frequency events would ideally involve simultaneous recordings of the same signal on both chip types. Unfortunately, due to inherent biological variability, both within the expression of the same event and across different events from separate slices, this is extremely challenging to achieve in practice.

Minor points

1. Line 113 – replace killed with euthanized

2. Line 160 – define hiPSCs

3. Line 72 – change intricated to intricate

4. Line 139 – change neurons to neuron

5. Line 140 – change cortexes with cortices

6. Line 407 – fix grammar “which allows to detect neuronal membrane”

7. Line 422 – add “=” in Scale bar 40um.

All the minor points were amended in the text.

Reviewer 2

Dear editor,

Regarding the manuscript titled ‘Enhanced electrophysiological recordings in acute brain slices, spheroids, and organoids using 3D high-density multielectrode arrays’ by Mapelli et al. The paper is excellently written, very thorough in details and provides a report on an important progress in the field of microelectrode arrays, addressing multiple common limitations found in the most commonly used planar MEAs arrays.

This is achieved by using a 3D high density micro electrode array (3D HD-MEA), which has several advantages over planar arrays. The authors highlight that planar arrays, while important in several instances have some key drawbacks, including:

- Planar arrays can only record signals which mostly originate from the outer layers, limiting the access to the dynamics of cells inside any 3D structure.

- Due to hardware limitations, it is challenging to record from multiple sites over large areas.

- Due to the intrinsic nature of how cells attach to a planar array, it imposes several difficulties regarding nutrient delivery, oxygenation and waste diffusion to avoid necrosis in the core of the tissue.

The authors main claims regarding the advantages of 3D HD-MEA are the following:

- The 3D HD-MEA microneedles reached the inner layers of samples without damaging network integrity.

- The microchannel network between microneedles improved tissue vitality and chemical compound diffusion.

- Signal recording and stimulation efficiency proved higher with the 3D HD-MEA than with a planar MEA.

- 3D HD-MEA resolved the challenge of recording from brain spheroids as well as cortical and spinal organoids.

Overall, I am very satisfied with the quality of the manuscript, and I consider it to be of great value to the scientific community, particularly due to the ability of this new technology to address or make use of biological preparations which are closer to the ones found in in-vivo work and to balance the advantages of high-density MEA while minimizing the damage to the tissue. Tests were intuitive, well-reasoned and the plots are appropriate and clear, the supplementary material is adequate and expands upon points that are of great use for specialists in the field and all the references seem logical and relevant.

Having said that, I do have some points that I believe are worthy of being addressed, both major and minor. None of my concerns demand extra experiments, but rather a more detailed explanation of the reasoning behind certain techniques and their consequences in comparing different arrays. I have divided my concerns into major and minor issues. Please see a detailed explanation below:

We thank the Reviewer for the highly positive comments and for carefully addressing the manuscript. We appreciate that the efforts put into this work have been acknowledged. We addressed all the issues raised. A point-by-point reply to Reviewer's comments and suggestions is provided below.

Major issues

1) Lines 436-439, it reads ‘The slices were first placed on a planar HD-MEA chip and then transferred onto a 3D HD-MEA chip (or vice versa) and recorded for 3 minutes in each case. The 3D HD-MEA chip showed increased performance in detecting activity from the cerebellar Purkinje cell layer compared to the planar HD-MEA…’. One of the main claims of this manuscript is that the 3d HD-MEA has a marked improvement in signal recording capabilities over planar MEA. For this, the authors compare the recording quality of the same cerebellar brain slice in a planar MEA and a 3D HD-MEA. As described in the methods (lines 258-261), there some methodological differences between the two recording arrays, namely that the planar array makes use of a platinum anchor, and that the activity in the brain slices was recorded 5 minutes after the slice was positioned on the chip.

My main concern is that using a planar array, 5 minutes may not be sufficient for the brain slice to achieve a good adhesion to the contacts on the array, thus, decreasing the recording quality of the planar array.

My understanding is that an ideal time is between 10-30 minutes before starting any recording; 5 minutes is going towards the bare minimum and this depends on the quality of the brain slices. I take it that the authors did their due diligence to ensure the highest possible quality of brain slices, and while I don’t consider that further experiments are necessary, I would encourage the authors to comment on their reasoning behind the timing for the planar arrays as well as the tradeoffs that occur when transplanting the same brain slice into the 3D HD-MEA.

In a way, the fact that more units and the overall yield is increased in 3D HD-MEA is in further support of the authors main arguments; but on the other hand, a naïve counter argument could be that the mechanical stress that the slices are subjected to could make the very same brain slice more easily accessible for the ‘needles’ in the 3D HD-MEA, increasing the number of units recorded and the overall yield. Thus, making the comparison between the planar and 3D arrays more difficult to interpret, since its possible that the mechanical stress that the brain slice is subjected to, may actually help the microneedles to penetrate the brain slice.

This concern is not addressed in any of the supplementary materials, and thus I bring it to attention. I would recommend that the authors expand their reasoning on the methods portion of the manuscript, section 2.6 ‘Electrophysiological recordings’.

We thank the Reviewer for this useful comment. Indeed, a few minutes are required to obtain proper stability of the coupling. We measured it in our conditions using cerebellar slices placed on 5 planar and 5 3D HD-MEAs. After stabilizing the slice on the chip, we recorded spontaneous activity for 25 minutes, every 3 or 5 minutes. The number of channels detecting spontaneous firing was assessed using spike detection with a hard threshold set at -100µV and considering only channels showing a spike rate higher than 10Hz (to avoid poorly coupled channels and not healthy neurons). By doing so, we observed a slight increase in the number of active channels in the planar chip and a slight decrease in the same parameter in 3D HD-MEAs during the first 3-5 minutes after placing the slices on the chips. After 5 minutes, the

---

## [Decision Letter · Decision Letter 1]

9 Jul 2025

Enhanced electrophysiological recordings in acute brain slices, spheroids, and organoids using 3D high-density multielectrode arrays

PONE-D-25-08619R1

Dear Dr. Mapelli,

We’re pleased to inform you that your manuscript has been judged scientifically suitable for publication and will be formally accepted for publication once it meets all outstanding technical requirements.

Kind regards,

Gerrit Hilgen

Academic Editor

PLOS ONE

Additional Editor Comments (optional):

Reviewers' comments:

Reviewer's Responses to Questions

**Comments to the Author**

1. If the authors have adequately addressed your comments raised in a previous round of review and you feel that this manuscript is now acceptable for publication, you may indicate that here to bypass the “Comments to the Author” section, enter your conflict of interest statement in the “Confidential to Editor” section, and submit your "Accept" recommendation.

Reviewer #1: All comments have been addressed

2. Is the manuscript technically sound, and do the data support the conclusions?

Reviewer #1: Yes

3. Has the statistical analysis been performed appropriately and rigorously? 

Reviewer #1: Yes

4. Have the authors made all data underlying the findings in their manuscript fully available?

Reviewer #1: Yes

5. Is the manuscript presented in an intelligible fashion and written in standard English?

Reviewer #1: Yes

6. Review Comments to the Author

Reviewer #1: Concerns addressed adequately. I have a couple of minor points.

1) Authors use room temperate and RT (I think that’s what it stands for) interchangeably without ever defining it like, room temperature (RT).

Line 117: room temperature

Line 202: RT

2) Authors never define SNRe, they talk about what they are finding with the equation in line 236, but they don’t ever use the SNRe acronym

7. PLOS authors have the option to publish the peer review history of their article (what does this mean? ). If published, this will include your full peer review and any attached files.

**Do you want your identity to be public for this peer review?** For information about this choice, including consent withdrawal, please see our Privacy Policy .

Reviewer #1: **Yes: ** R Ryley Parrish

---

## [Editor Report · Acceptance letter]

PONE-D-25-08619R1

PLOS ONE

Dear Dr. Mapelli,

I'm pleased to inform you that your manuscript has been deemed suitable for publication in PLOS ONE. Congratulations! Your manuscript is now being handed over to our production team.

Kind regards,

on behalf of

Dr. Gerrit Hilgen

Academic Editor

PLOS ONE